# Defining cell type-specific immune responses in a mouse model of allergic contact dermatitis by single-cell transcriptomics

Youxi Liu[1†], Meimei Yin[1†], Xiaoting Mao[2†], Shuai Wu[3†], Shuangping Wei[2,4], Shujun Heng[1], Yichun Yang[1], Jinwen Huang[5], Zhuolin Guo[6], Chuan Li[2], Chao Ji[5], Liu Hu[2*], Wenjie Liu[1*], Ling-juan Zhang[1*]

[1]State Key Laboratory of Cellular Stress Biology, School of Pharmaceutical Sciences, Xiamen University, Xiamen, China; [2]Zhejiang Yangshengtang Institute of Natural Medication Co Ltd, Hangzhou, China; [3]State Key Laboratory of Cellular Stress Biology, School of Life Sciences, Xiamen University, Xiamen, China; [4]Yang Sheng Tang (Anji) Cosmetics Co Ltd, Zhejiang, China; [5]Department of Dermatology, The First Affiliated Hospital of Fujian Medical University, Fuzhou, China; [6]Department of Dermatology, Shanghai Tenth People's Hospital, Tongji University School of Medicine, Shanghai, China

**\*For correspondence:**
lhu@mail.yst.com.cn (LH);
wjliu@xmu.edu.cn (WL);
lingjuan.zhang@xmu.edu.cn
(L-jZ)

[†]These authors contributed equally to this work

**Abstract** Allergic contact dermatitis (ACD), a prevalent inflammatory skin disease, is elicited upon repeated skin contact with protein-reactive chemicals through a complex and poorly characterized cellular network between immune cells and skin resident cells. Here, single-cell transcriptomic analysis of the murine hapten-elicited model of ACD reveals that upon elicitation of ACD, infiltrated CD4[+] or CD8[+] lymphocytes were primarily the IFNγ-producing type 1 central memory phenotype. In contrast, type 2 cytokines (IL4 and IL13) were dominantly expressed by basophils, IL17A was primarily expressed by δγ T cells, and IL1β was identified as the primary cytokine expressed by activated neutrophils/monocytes and macrophages. Furthermore, analysis of skin resident cells identified a sub-cluster of dermal fibroblasts with preadipocyte signature as a prominent target for IFNγ[+] lymphocytes and dermal source for key T cell chemokines CXCL9/10. IFNγ treatment shifted dermal fibroblasts from collagen-producing to CXCL9/10-producing, which promoted T cell polarization toward the type-1 phenotype through a CXCR3-dependent mechanism. Furthermore, targeted deletion of *Ifngr1* in dermal fibroblasts in mice reduced *Cxcl9/10* expression, dermal infiltration of CD8[+] T cell, and alleviated ACD inflammation in mice. Finally, we showed that IFNγ[+] CD8[+] T cells and CXCL10-producing dermal fibroblasts co-enriched in the dermis of human ACD skin. Together, our results define the cell type-specific immune responses in ACD, and recognize an indispensable role of dermal fibroblasts in shaping the development of type-1 skin inflammation through the IFNGR-CXCR3 signaling circuit during ACD pathogenesis.

## eLife assessment

This **important** study uses single-cell RNA-seq to obtain a more granular understanding of cell subsets within allergic contact dermatitis in a model system with DNFB. The **convincing** data revela unique subpopulations of dermal fibroblasts as key responders to interferon gamma and likely as mediators of dermatitis. This study has many novel aspects and provides a unique resource as well.

## Introduction

Allergic contact dermatitis (ACD), one of the most common skin inflammatory diseases, is affecting 15–20% of the world population (*Vocanson et al., 2009*). It is a delayed-type hypersensitivity reaction triggered by repeated skin contact with protein-reactive chemicals with low molecular weight, such as haptens. A typical hapten-induced ACD reaction involves a sensitization phase, followed by an elicitation phase. In the sensitization phase, high doses of low-molecular-weight haptens (<500 Da) penetrate through the skin stratum corneum and are captured by Langerhans cells and dendritic cells, which process these antigens and present them to naïve T cells in the skin-draining lymph nodes, priming and clonally expanding the hapten-specific T cells systemically (*Kaplan et al., 2012*). In the elicitation phase, skin re-exposure to haptens (even at low doses) triggers the activation of hapten-specific T cells and a cascade of inflammatory reactions, generally, 24–72 hr after exposure (*Manresa, 2021*).

Hapten-triggered innate immune response involves the release of a panel of proinflammatory cytokines and/or chemokines, such as IL1β, TNFα, CXCL9/10, CCL17, and CCL20, which contribute to T cell trafficking to the skin (*Kaplan et al., 2012*; *Meller et al., 2007*; *Dufour et al., 2002*). Most of the T cells are recruited by chemokines in an antigen-non-specific manner, but when the recruited T cells encounter specific antigen, T cells proliferate and are activated locally, initiating the inflammatory cascade and promoting the influx and/or activation of cytotoxic cells (CD8[+] T cells and natural killer T cells), mast cells (MCs) and M1-polarized inflammatory macrophages (MACs) (*Vocanson et al., 2009*; *Dufour et al., 2002*; *Chai et al., 2022*). Both CD4[+] and CD8[+] T cells are involved in inflammatory responses, and CD8[+] T cells are considered the main effector cells (*Chai et al., 2022*; *Vocanson et al., 2006*). Recruited lymphocytes secrete a panel of inflammatory cytokines, such as IFNγ, IL4, IL13 and IL17, which act on skin-resident cells such as keratinocytes, upregulating the expression of adhesion molecules and cytokines/chemokines, thereby recruiting more T cells, macrophages, mast cells to the affected site (*Kaplan et al., 2012*). However, the cell-type-specific immune response driving ACD pathology still remains poorly defined.

Dermal fibroblasts (dFBs), the main resident cell type in skin dermis, are highly heterogeneous, and can be classified based on their anatomical location (*Huang et al., 2022*). By single-cell RNA-seq (scRNA-seq), we have recently classified dFBs into distinct non-adipogenic and adipogenic subgroups, and defined several dermal adipocyte lineage cells (*Sun et al., 2023*). The primary function of dFBs is to provide structural support for the skin by producing extracellular matrix (ECM) molecules, such as collagen and proteoglycans (*Huang et al., 2022*), and emerging new studies have recognized that dFBs also have important innate immune functions. We have shown that adipogenic dFBs fight against bacterial infections by producing antimicrobial peptides (*Zhang et al., 2021*; *Zhang et al., 2019*; *Zhang et al., 2015*). In addition, a recent study identified a dFB subset that is involved in activating cytotoxic CD8[+] T cells during vitiligo pathogenesis (*Xu et al., 2022*). However, how dFBs are involved in the inflammatory circuit with lymphocytes during ACD pathogenesis remains largely unexplored.

In this study, ACD was induced in mice by sequential application of 2,4-dinitrofluorobenzene (DNFB), and this is the most commonly used murine model for ACD (*Manresa, 2021*; *Karsak et al., 2007*; *Miyake et al., 2022*). In-depth scRNA-seq analysis was performed to characterize of how cytokines associated with ACD were differentially expressed by various immune or skin resident cells. In addition to immune cells, we performed in-depth analysis of the immune response of dFBs. Furthermore, the interaction between dFBs and T cells was investigated by in vitro and in vivo approaches. Finally, human ACD skin samples were analyzed to validate the clinical relevance of our observations in mice. Results from this study provide a better understanding of the cell-type specific immune responses underlying ACD pathogenesis, and unravel the previously unknown role of dFBs in shaping the type 1 immune response in ACD.

## Results

### Characterization of mouse allergic skin immune response by single cell RNA transcriptomic analysis

Mice were first sensitized to hapten by applying high dose of DNFB to dorsal skin, and 6 days after sensitization, ACD was elicited by applying a low dose of DNFB to the ears (*Figure 1A*; *Manresa,*

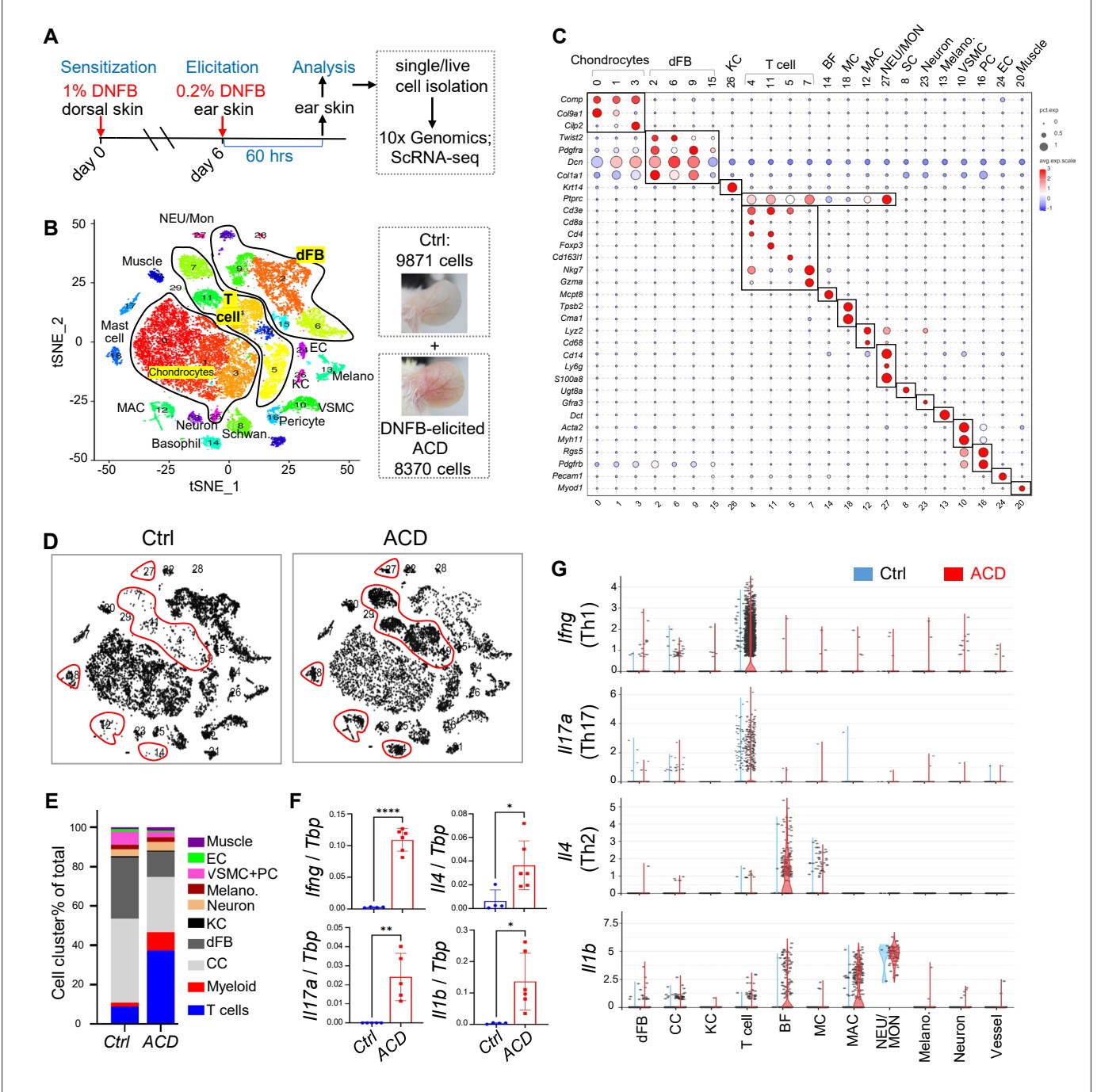

**Figure 1.** Characterization of mouse allergic skin immune response by single cell RNA transcriptomic analysis. (**A**) Overview of the experimental setting. Allergic contact dermatitis (ACD)-like skin inflammation was triggered by sequential sensitization and elicitation of ACD by topical application of 1% or 0.2% DNFB on dorsal or ear skin as indicated. Ear skin was collected at 60 hr post-elicitation for analyses. (**B**) tSNE plots showing the distribution of various cell clusters marked by a color code. (**C**) Bubble plots showing the expression of marker genes for each cell cluster. Abbreviations: *dFB*, dermal fibroblast; *MC*, mast cell; *MAC*, macrophage; *BF*, basophil; *NEU*, neutrophil; *Mon*, monocytes; *VSMC*, vascular smooth muscle cell; *PC*, pericytes; *KC*, keratinocyte; *EC*, endothelial cell; *SC*, Schwann cell. (**D**) tSNE plots showing how cells were differentially distributed in control and ACD skin samples. Red lines circle key immune cell populations. (**E**) Stacked bar graph showing the percentage of each cell cluster in the control and ACD samples. (**F**) qRT-PCR analysis showing the expression of indicated genes (n=4–6/group). (**G**) Violin plots showing the expression of indicated genes in the control and ACD samples.

The online version of this article includes the following figure supplement(s) for figure 1:

**Figure supplement 1.** Establishment of the DNFB-elicited ACD mouse model.

*2021*). The development of skin erythema and thickening peaked at ~24 hr and lasted until ~60 hr post-treatment (*p.t.*) (*Figure 1—figure supplement 1A and B*). Histological analyses showed that the skin dermis drastically expanded at 24 hr *p.t.*, and the dermis became heavily infiltrated with mononuclear cells by 60 hr *p.t.* (*Figure 1—figure supplement 1C*).

To characterize the cell type-specific immune responses in this DNFB-elicited mouse ACD model, we performed scRNA-seq of control and ACD ear skin samples, and identified up to 30 cell clusters (*Figure 1B*), which were grouped into several cell types, including chondrocytes, dermal fibroblasts, keratinocytes, T cells, myeloid cells (basophils/BFs, mast cells/MCs, macrophages/MACs, neutrophils/NEUs, monocytes/Mon) and other skin resident cells based on marker gene expression (*Figure 1C*, *Figure 1—figure supplement 1D*, and *Supplementary file 2*). Note that because cluster 27 expressed both neutrophil marker *Ly6g* (*Lee et al., 2013*) and monocyte marker *Cd14* (*Antal-Szalmas et al., 1997*), we named this cluster 'NEU/Mon'. Furthermore, differential cell distribution plots (*Figure 1D*) showed that lymphocytes and myeloid cells were the main cell types recruited to the skin after elicitation of ACD (*Figure 1E*).

ACD is an inflammatory skin disease, driven by a complicated cell-cell network mediated by several T cell-associated cytokines, including Th1 cytokine IFNγ, Th2 cytokines IL4, IL13, and Th17 cytokine IL17 (*Howie et al., 1996*). In addition, IL10 participates in skewing of the Th2 response in a murine model of ACD (*Laouini et al., 2003*), and IL1β is induced in ACD and potentiates the immune response from both T cells and resident cells (*Terui et al., 2021*). First, we confirmed that the expression of these cytokines was significantly elevated in ACD compared to that in the control skin samples (*Figure 1F* and *Figure 1—figure supplement 1E and F*). scRNA-seq gene expression plots revealed that *Ifng*, *Il17*, and *Il10* were primarily expressed by T cell sub-clusters as expected; to our surprise, *Il4* and *Il13* were primarily produced by basophils, and *Il1b* was detected at the highest levels in the neutrophil/monocyte cluster and at intermediate levels in other myeloid cells, including basophils and macrophages (*Figure 1G*, *Figure 1—figure supplement 1G*). Furthermore, we found that the expression of *Ifng*, but not *Il4* or *Il17a*, was rapidly induced in skin draining lymph nodes at 24 hr after ACD elicitation (*Figure 1—figure supplement 1H*). This suggests a robust and systemic activation of type 1 memory T cell response in the early stage of ACD, and the migration of these lymphatic memory T cells to the skin may contribute to the exacerbation of skin inflammation.

## T cells are primarily polarized to the IFNγ-producing type-1 inflammatory phenotype in ACD

To further define the T cell-specific immune response in ACD, T cell clusters were re-grouped into eight clusters (r0~r7; *Figure 2A*), which were defined as CD8[+], CD4[+], Treg, δγT, NKT, NK/ILC1, and ILC2, based on marker gene expression (*Figure 2—figure supplement 1A and B*). ACD led to a notable increase in the percentage of CD8[+], CD4[+], Treg, and NK/ILC1, but not in that of δγT and ILC2 cells (*Figure 2B*). Antigen-specific CD8 or CD4 memory T cells can be classified into CD62[hi]/CCR7[hi]/CD28[hi]/CD27[hi]/CX3CR1[lo] central memory T cells (Tcm), CX3CR1[hi]/Cd28[hi]/Cd27[lo]/CD62[lo]/CCR7[lo] effector memory T cells (Tem), and CD49a[hi]/CD103[hi]/ CD69[hi]/BLIMP1[hi] tissue-resident memory T cells (Trm) (*Cheon et al., 2023*; *Martin and Badovinac, 2018*; *Benichou et al., 2017*; *Park et al., 2023*; *Mackay et al., 2013*). We observed that in ACD skin, CD4[+] and CD8[+] T cells predominantly expressed marker genes associated with Tcm including *Cd28*, *Cd27*, *Ccr7*, and *S1pr1/Cd62l*. In contrast, marker genes associated with Tem (*Cx3cr1*) and Trm (*Itga1/Cd49a*, *Itgae/Cd103*, *Cd69* and *Prdm1/Blimp1*, *Cd127/Il7r*) were only scarcely expressed in these αβ T cells (*Figure 2—figure supplement 1C–D*). These results suggest that ACD predominantly triggers a central memory T cell response in the skin.

Furthermore, gene expression plots (*Figure 2C–D* and *Figure 2—figure supplement 1C*) showed that *Ifng* was robustly induced in CD4[+], CD8[+], and NK/ILC1 T cell clusters, whereas *Il10* and *Il17a* were moderately induced only in the Treg and δγT (r5) cell clusters, respectively. In contrast, *Il4* was barely detectable in all T cell subsets. The expression of *Ifng*, *Il17a*, and *Il10* aligned well with that of key T cell lineage transcription factors (TFs), including *Tbx21* (for Th1), *Rorc* (for Th17), and *Foxp3* (for Treg), respectively (*Figure 2—figure supplement 1E and F*).

GO pathway analysis of CD4[+] and CD8[+] T cell clusters revealed that chemokine signaling and type II interferon signaling pathways were among the top upregulated, and TGFβ and PI3K-Akt-mTOR signaling were among the top downregulated pathways in ACD compared control (*Figure 2E*). Volcano plot of differentially expressed genes (*Figure 2F*) showed that *Ifng* and several IFN-JAK-STAT-related

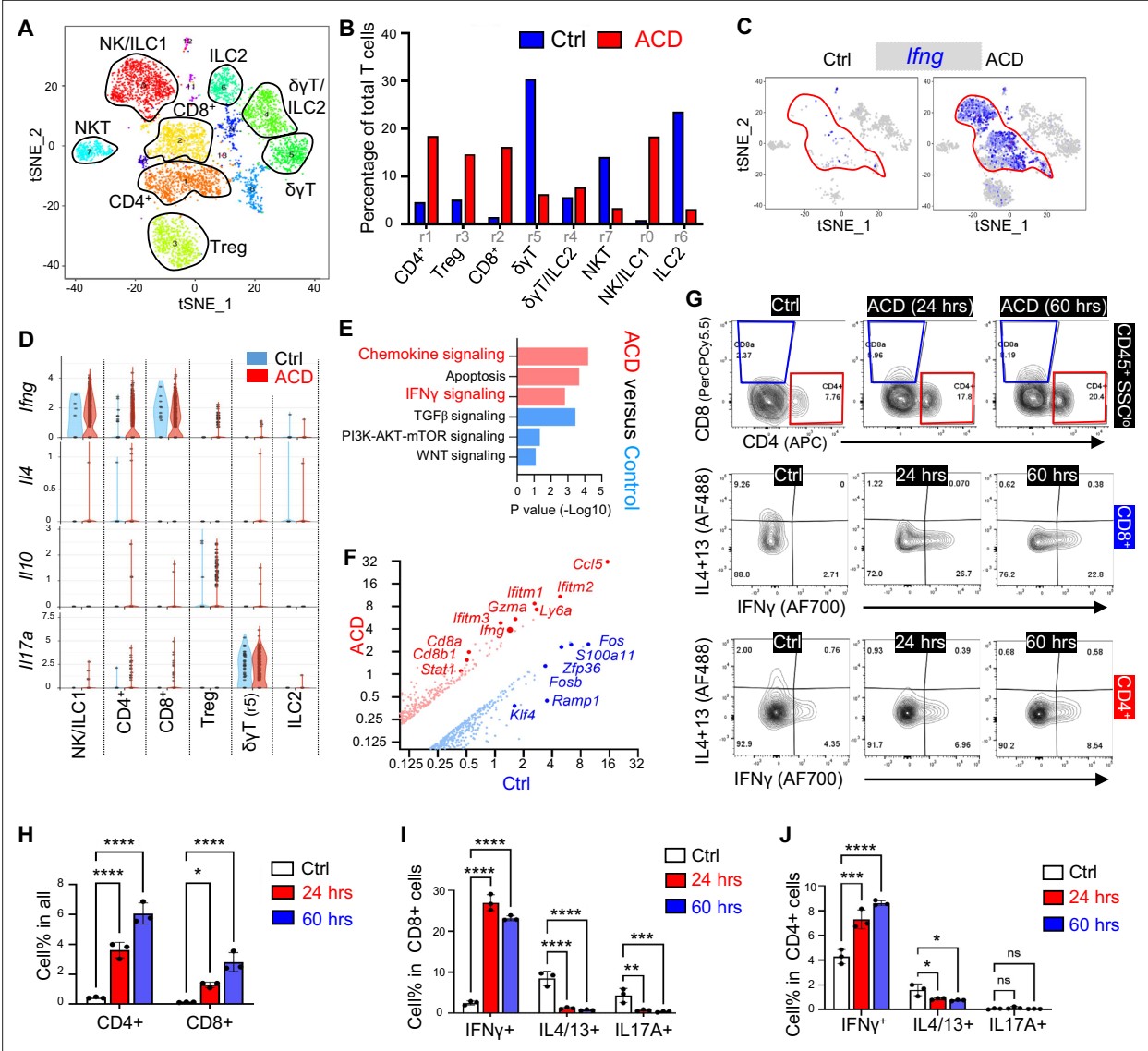

**Figure 2.** T cells are primarily polarized to the IFNγ-producing type-1 inflammatory phenotype in ACD. (**A**) tSNE plots showing cell distribution of the CD45+CD3+ or THY1+ T cell population after re-clustering. (**B**) Bar graphs showing the percentage of each T cell sub-cluster in the control and the ACD samples. (**C**) tSNE plots showing cell distribution or the expression of *Ifng* in the control and the ACD samples. (**D**) By sample violin plots showing the expression of indicated genes across various T cell sub-clusters. (**E**) WIKI pathway analysis showing the top upregulated (red) or downregulated (blue) pathways in CD4+ and CD8+ T cells in ACD compared to control skin. (**F**) Volcano plot showing differentially expressed genes in control and ACD samples within the CD4+ and CD8+ T cells. (**G–J**) FACS plots and/or quantified bar graphs showing the percentage of CD4+ or CD8+ T cells (upper panel in G), or the percentage of IFNγ+IL4/13-, IFNγ-IL4/13+, or IFNγ-IL17A+ cells in CD8+ (middle panel in G) and CD4+ (lower panel in G) T cells in control, ACD (24 hr) and ACD (60 hr) ear skin samples (n=3/group). All error bars indicate mean ± SEM. *p<0.05, **p<0.01, ***p<0.001, ****p<0.0001, ns, non-significant.

The online version of this article includes the following figure supplement(s) for figure 2:

**Figure supplement 1.** Characterization of T cell activation in ACD.

genes (including *Ifitm1/2/3* and *Stat1*) were among the top upregulated genes in ACD, and genes with reported inhibitory functions for T cell activation and/or Th1 polarization, including *Fos* (**Xiao et al., 2012**), *Zfp36* (**Moore et al., 2018**), and *Klf4* (**An et al., 2011**), were among the top downregulated genes in ACD, compared to those in the control.

Next, we aimed to validate the expression of IFNγ in CD4+ and CD8+ T cells in ACD. First, a time-dependent increase in *Cd4, Cd8* and *Ifng* transcript levels was observed in the DNFB-elicited ACD skin (**Figure 2—figure supplement 1G**). FACS analysis (**Figure 2G** and **Figure 2—figure supplement**

*1H–K*) showed that DNFB-elicited ACD resulted in a rapid increase in both CD4$^+$ and CD8$^+$ T cells, in which IFNγ expression was significantly induced, whereas IL4/IL13 and/or IL17 expression was inhibited. Together, these results indicate that T cells are primarily polarized to the IFNγ-producing type-1 inflammatory phenotype in DNFB-elicited ACD.

## Basophils are major source of type-2 cytokines in ACD

scRNA-seq analysis of myeloid cell clusters showed that ACD led to an increase of basophils, mast cells, neutrophils, and macrophages (*Figure 3A*). Violin plot analyses of cytokines across major skin or immune cell types confirmed that *Il4* was primarily expressed by basophils and mast cells, whereas *Il1b* was primarily expressed by neutrophils and macrophages (*Figure 3B* and *Figure 3—figure supplement 1A*).

Basophils and mast cells have similar functions and developmental processes, and both cell types express the high-affinity Fc receptor for IgE (FceR1a), the activation of which triggers degranulation and release of proteases, histamine, and a panel of proinflammatory cytokines (*Hamey et al., 2021*). Volcano plots of genes expressed by mast cells (*Cd11b⁻Cma1⁺Fcer1a⁺*) and basophils (*Cd11b⁺Mcpt8⁺Fcer1a⁺*) showed that, in comparison, basophils expressed higher levels of cytokines/chemokines, such as *Il4*, *Il13*, and *Il6*, and mast cells expressed higher levels of *Ccl2* and *Ccl7* (*Figure 3C*), revealing their differential immune responses in ACD.

Next, we aimed to validate the involvement of BFs and MCs in the DNFB-elicited type 2 immune response. First, the expression levels of *Mcpt8* (BF marker), but not *Cma1* (MC marker), were elevated in a time-dependent manner upon ACD-elicitation (*Figure 3D*). FACS analysis of BFs and MCs, based on the surface expression of CD11B and FcεR1a (*Figure 3E and F* and *Figure 3—figure supplement 1B*), showed that ACD triggered a rapid increase in the percentage of IL4/IL13-producing CD11B$^+$ FcεR1a$^+$ BFs as early as 24 hr post-elicitation. In contrast, there was only a moderate and delayed increase in IL4/IL13-producing CD11B$^-$ FcεR1a$^+$ MCs after ACD-elicitation (*Figure 3E and F*). In addition, immunostaining analysis showed that elicitation of ACD led to dermal infiltration of both FcεR1a$^+$ TPSB$^-$ BFs and FcεR1a$^+$ TPSB$^+$ MCs, in which Th2 cytokines were primarily co-localized (*Figure 3—figure supplement 1C*).

## Characterization of the immune responses of neutrophils, monocytes and macrophages in ACD

Time-dependent induction of *Il1b* and recruitment of CD68$^+$ MAC and Ly6G$^+$ NEU after ACD elicitation was confirmed by qRT-PCR analysis (*Figure 3G*). Sc-RNAseq revealed that *S100a8/9*, *Il1b*, and *Cxcl2* were among the most highly expressed genes in the neutrophil/monocyte cell cluster (*Figure 3—figure supplement 1D*), and *Il1b* was among the top inducible genes in *Cd68$^+$* + after ACD elicitation (*Figure 3H*). Interestingly, genes associated with immunosuppressive or M2 MACs (*Cd74*, *H2-Ab*, and *Ccl24*) were among the most downregulated genes in MACs after ACD elicitation (*Figure 3H*). Furthermore, GO pathway analysis identified 'cellular response to IFNγ' as the top upregulated pathway, and 'lipid metabolic' and 'actin cytoskeleton organization' as the top downregulated pathways in MACs after elicitation of ACD (*Figure 3I*). Anti-inflammatory function of M2 MACs is fueled by fatty acid oxidation (*Batista-Gonzalez et al., 2019*), and changes in actin organization and cell shape are hallmarks of MAC activation (*McWhorter et al., 2013*). These results provide insights into the mechanisms underlying myeloid activation in ACD.

FACS analysis of NEUs, monocytes, and MACs based on protocol reported by *Rahman et al., 2017* confirmed that elicitation of ACD led to a time-dependent increase in NEU recruitment and a shift of MACs from a resting (Ly6C$^{lo}$) to an inflammatory (Ly6C$^{hi}$) state (*Figure 3J–L* and *Figure 3—figure supplement 1E and F*). In contrast, elicitation of ACD led to a transient influx of monocytes peaking at 24 hr post treatment (*Figure 3M*, *Figure 3—figure supplement 1G*). Together, these data suggest that IL1β may be a key effector molecule executing the proinflammatory function of NEUs, MONs and MACs, and that IFNγ may play an upstream role in activating MACs during ACD pathogenesis.

## Characterization of the immune response of dermal fibroblasts

Although dermal fibroblasts are the main resident cell type in the skin dermis (*Huang et al., 2022*), their role in regulating ACD remains largely unexplored. PDGFRA, a well-characterized skin pan-fibroblast marker (*Lynch and Watt, 2018*; *Driskell et al., 2013*), was used to identified dFBs, and

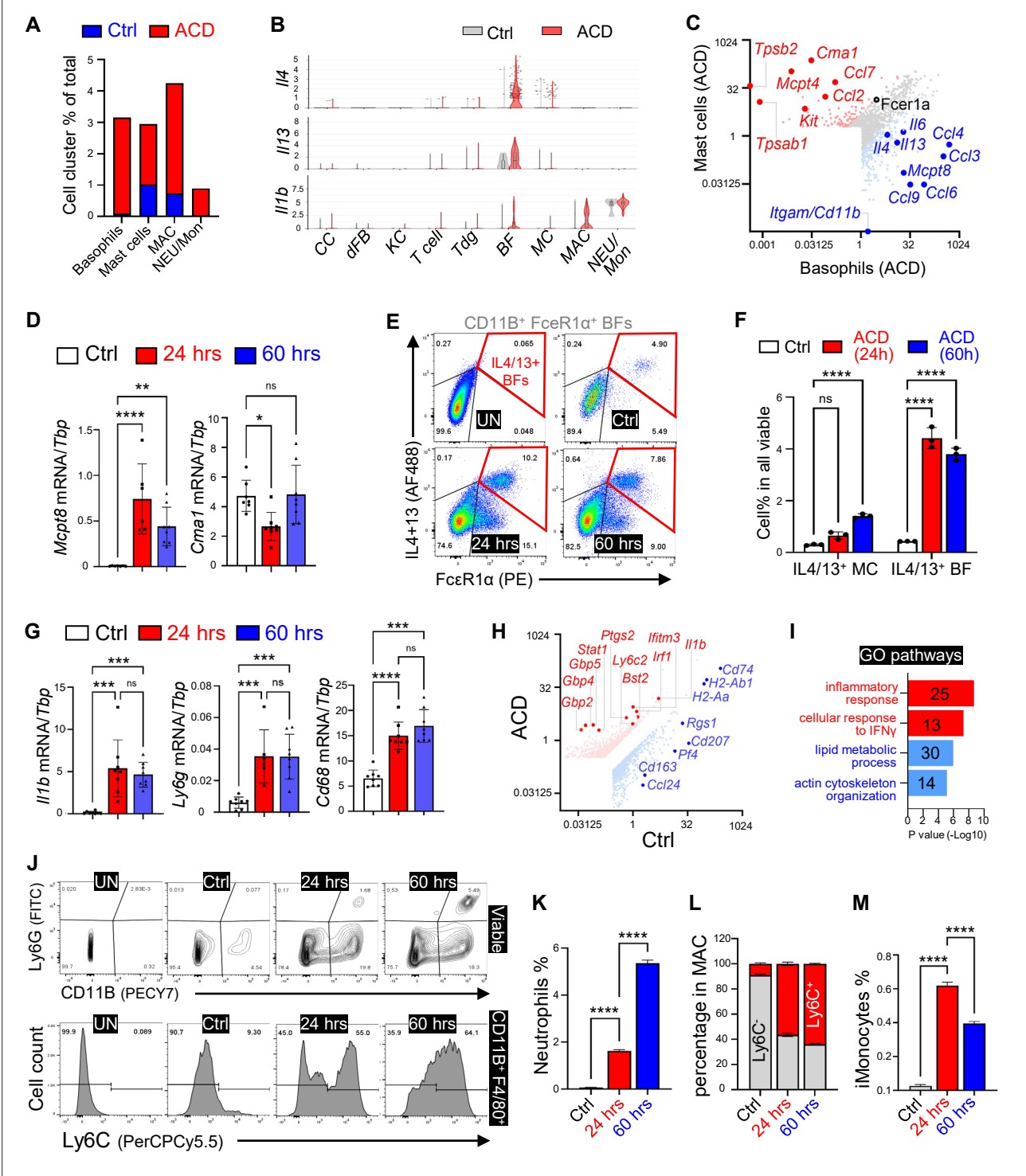

**Figure 3.** Characterization of myeloid cell activation in ACD. (**A**) Stacked bar graphs showing the percentage of indicated myeloid cell populations in the control and the ACD samples. (**B**) Violin plots showing the expression of indicated genes across major skin resident and immune cells as indicated. (**C**) Gene expression plot showing differentially expressed genes in basophils (BF, blue dots) and mast cells (MC, red dots) within the ACD sample. (**D**) qRT-PCR analysis of indicated genes in ear skin samples (n=6~8/group). (**E–F**) FACS plots (**E**) and quantified bar graphs (**F**) showing the percentage of IL4/13+ mast cells or basophils within all viable cells in ear skin samples (n=3/group). (**G**) qRT-PCR analysis of indicated genes in ear skin samples (n=6~8/group). (**H**) Gene expression plot showing differentially expressed genes in control and ACD samples within the macrophage cell cluster. (**I**) GO pathway analysis of the upregulated genes (red) or downregulated genes (blue) in macrophage population after ACD elicitation. (**J–M**) FACS plots (**J**) and quantified bar graphs (**K–M**) (n=3/group) showing the percentage of CD11B+Ly6G+ neutrophils (NEU) in all cells (**K**), the percentage of Ly6C+ or

*Figure 3 continued on next page*

*Figure 3 continued*

Ly6C⁻ cells within CD11B⁺F4/80⁺ macrophages (**L**), or the percentage of inflammatory monocytes (**M**) (see FACS plots in **Figure 3—figure supplement 1G**), near skin samples. Unstained (UN) plot was shown as negative gating control. All error bars indicate mean ± SEM. *p<0.05, **p<0.01, ***p<0.001, ****p<0.0001, ns, non-significant.

The online version of this article includes the following figure supplement(s) for figure 3:

**Figure supplement 1.** Analysis of myeloid cell activation in ACD.

*Pdgfra*⁺ dFBs were reclustered into seven sub-clusters (r0 to r6) (**Figure 4A**). According to dFB and/or adipocyte-lineage cell marker gene profiles reported by our or others' work (**Sun et al., 2023**; **Driskell et al., 2013**; **Plikus et al., 2021**; **Joost et al., 2020**; **Zachara et al., 2022**; **Schwalie et al., 2018**), the dFB sub-clusters are defined as *Mfap5⁺Dlk1⁺Ly6a⁺*adipocyte progenitors (AP in r2), *Fmo2⁺Ly6a⁺*adipose regulatory cells (Areg in r6), *Ly6a⁺Lpl⁺Apoe⁺Icam1⁺*preadipocytes (pAd, in r1 and r5), *Ly6a^{lo}* or *Lrig1⁺* + or papillary dFBs (RET/PAP in r0), and *Ly6a^{lo}Trps1^{hi}* peri-follicular dFBs (including dermal papilla cells, r4; **Figure 4B**, **Figure 4—figure supplement 1A**).

Bubble plots and violin plots showed that ACD elicitation led to an increase of the expression of a panel of IFN-inducible genes (*Cxcl9, Cxcl10, Stat1*), accompanied with a loss of the expression of pAd signature genes (*Lpl, Apoe*) and a panel of ECM genes (*Col1a1, Col3a1*) in the r5 pAd cluster (**Figure 4C** and **Figure 4—figure supplement 1B** and C). In addition, scenic TF analysis revealed that the dFB_r5 cluster was specifically enriched with several TFs downstream of the interferon pathway (*Stat1, Irf1,* and *Irf7*; **Figure 4D**). Furthermore, GO pathway analysis of dFB_r5 cells showed that pathways including response to virus, cellular response to IFNγ, and positive regulation of T-cell-mediated cytotoxicity were among the top upregulated pathways, whereas ECM organization was among the top downregulated pathways in ACD compared to control (**Figure 4E**). Accordingly, volcano gene expression plots of dFB_r5 cluster showed that *Cxcl9, Cxcl10, Ccl2, Stat1,* and other antiviral genes were the top upregulated genes, whereas ECM genes were the top downregulated genes in ACD compared to control (**Figure 4F**).

Violin plots of major skin and immune cell types showed that inflammatory genes related to the IFNGR pathway (*Stat1, Cxcl9, Cxcl10*) were expressed at highest levels in the dFB_r5 cluster, whereas *Ifng* and *Cxcr3* (receptor for CXCL9/10 ligands) was mainly expressed by T cells (**Figure 4G** and **Figure 4—figure supplement 1D**). qRT-PCR validated the time-dependent induction of chemokines after ACD elicitation (**Figure 4—figure supplement 1E**). Co-immunostaining analyses showed that CXCL9⁺ or CXCL10⁺ cells co-expressed PDGFRA in the lower dermis where PLIN1⁺ adipocytes are enriched of ACD skin samples (**Figure 4H–J** and **Figure 4—figure supplement 1F–H**), supporting that dFB is the major cell source for CXCL9/10 in ACD. In addition, phosphor-STAT1 (pSTAT1), a key signaling molecule activated by IFNγ, was detected primarily in PDGFRA⁺Ly6A⁺ pAds located within the lower dermis (**Figure 4—figure supplement 1I and J**). Furthermore, in ACD skin dermis, CXCL9 expression largely co-localized with ICAM1 (**Figure 4—figure supplement 1K**), a marker for committed pAds (**Merrick et al., 2019**). This further confirms that CXCL9 is specifically induced in the pAd subset of dFBs. Together, we have identified a cluster of dFB with pAd signature as a cellular source of T cell chemokines CXCL9/10 in ACD, suggesting that dFBs may play a role in the development of type-1 immune response by interacting with T cells.

## Interaction between dermal fibroblasts and T cells via the IFNγ-CXCL10-CXCR3 signaling axis

A subset of IFNγ-responsive dFBs has been identified to recruit and activate CD8⁺ cytotoxic T cells during the pathogenesis of vitiligo (**Xu et al., 2022**). By comparing these active vitiligo mouse dFBs with the active ACD pAds (dFB_r5), we found that these two cell populations are enriched with highly similar IFNγ-inducible genes (**Figure 5—figure supplement 1A** and **Supplementary file 3**), suggesting that the IFNγ-responsive pAds may also contribute to the activation of type 1T cells in the context of ACD.

We next aimed to validate the interaction between dFBs and T cells. First, violin plots showed that among all other dFB clusters, the *Pdgfra⁺Ly6A⁺Icam1⁺*dFB_r5 pAds exhibited the highest expression levels of *Ifngr1* and *Ifngr2*, along with *Stat1, Cxcl9,* and *Cxcl10* (**Figure 5—figure supplement 1B**). IFNGR1 plays a dominant role in ligand binding, while IFNGR2 is primarily responsible for activating

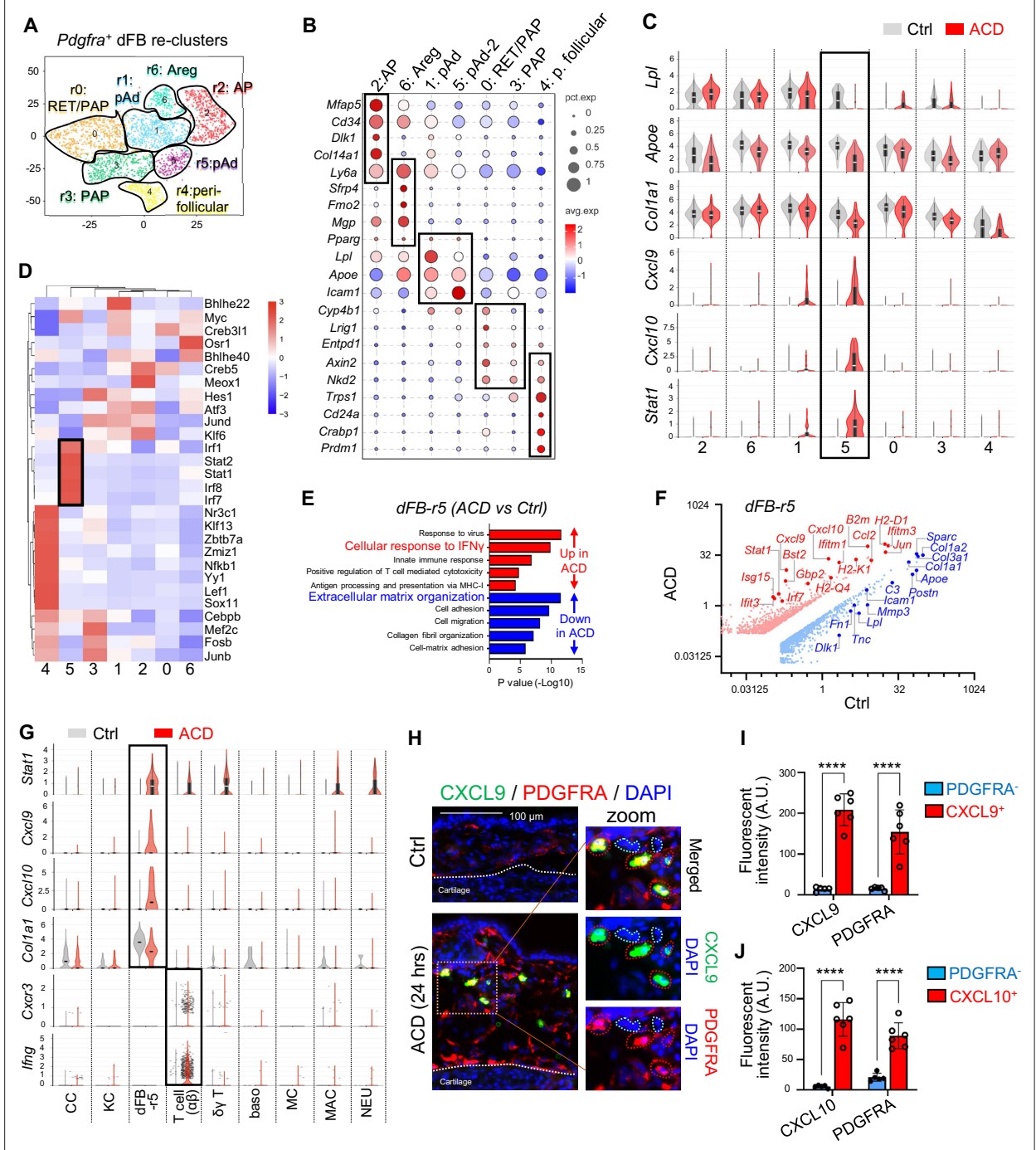

**Figure 4.** Characterization of the immune response of dermal fibroblasts in ACD. (**A**) tSNE plots showing cell distribution of the Pdgfra[+] dermal fibroblasts after re-clustering. (**B**) Bubble plots showing the expression of marker genes for each dFB cell cluster. Abbreviations: AP, adipocyte progenitors; Areg, adipogenesis-regulatory cells; pAd, preadipocytes; RET/PAP, reticular and/or papillary dFBs; pF, peri-follicular dFBs. (**C**) Violin plots showing the expression of indicated genes across various dFB sub-populations in the control and the ACD samples. (**D**) SCENIC analysis showing the top enriched transcriptional factors in various dFB clusters. (**E**) GO pathway analysis of the upregulated genes (red) or downregulated genes (blue) in the r5 dFB cluster after ACD elicitation. (**F**) Volcano plot showing differentially expressed genes in control and ACD samples within the r5 dFB cluster. (**G**) Violin plots showing the expression of indicated genes across various cell populations in the control and the ACD samples. (**H**) Frozen sections of control and ACD ear skin samples were subjected to immunostaining analysis using antibodies against CXCL9 (green) and PDGFRA (red). Nuclei were counter stained by DAPI (blue). Dermal CXCL9[+] or PDGFRA[-] cells were highlighted by either red- or green-dotted lines in the zoom-in panel. Scale bar, 200 μm. Zoom-in image is shown on the right-hand side. (**I–J**) Quantified results showing the fluorescent intensity (arbitrary unit, AU) of CXCL9, CXCL10

*Figure 4 continued on next page*

*Figure 4 continued*

or PDGFRA in the dermal CXCL9/10⁺ cells or PDGFRA⁻ cells shown in (**H**) or *Figure 4—figure supplement 1G* (n=5~6/group). All error bars indicate mean ± SEM. *p<0.05, ***p<0.001, ****p<0.0001, ns, non-significant.

The online version of this article includes the following figure supplement(s) for figure 4:

**Figure supplement 1.** Characterization of the immune response of dermal fibroblasts in ACD.

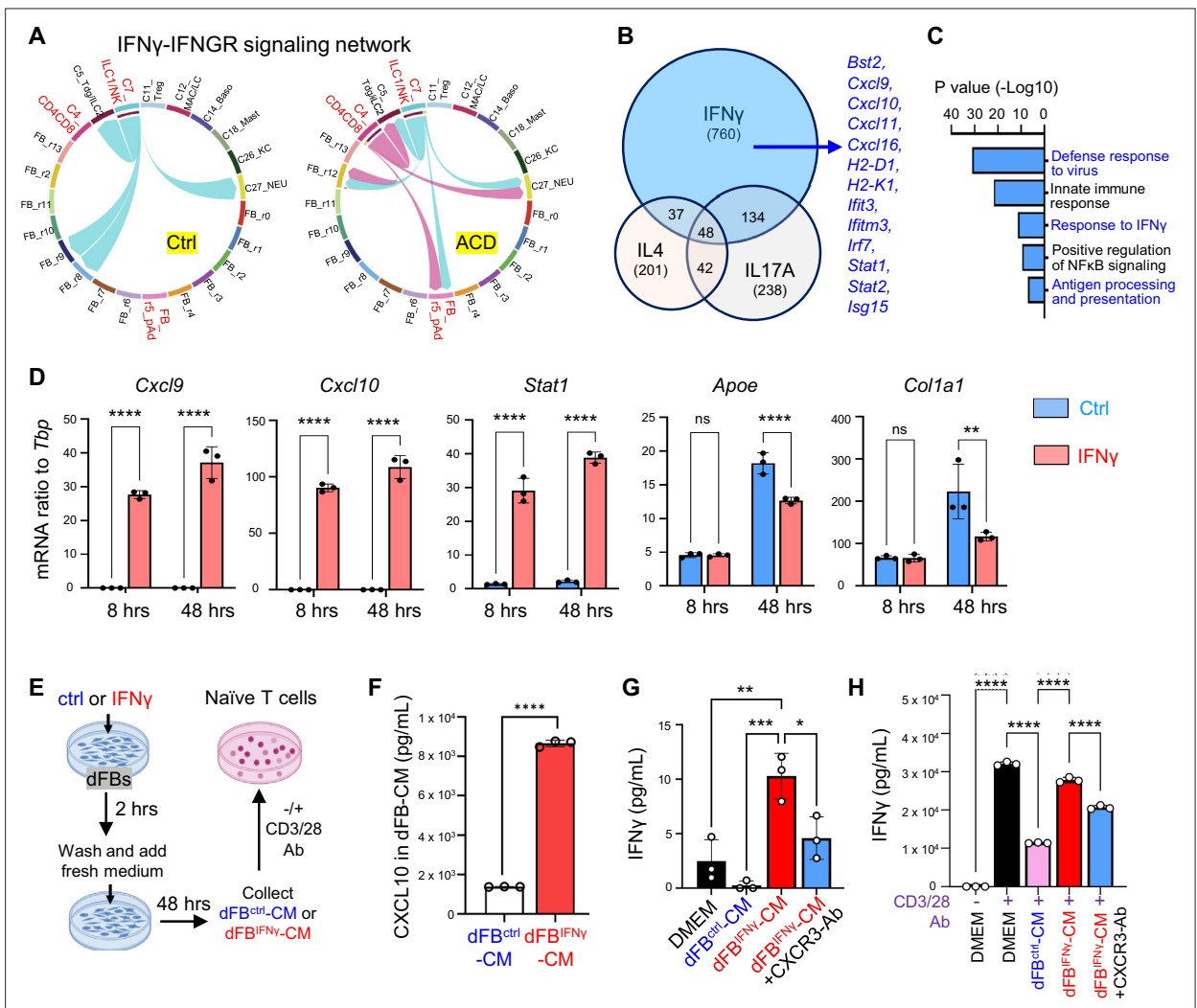

**Figure 5.** Interaction between dFBs and T cells via the IFNG-CXCL10-CXCR3 signaling axis in ACD. (**A**) Circle plot from cell-chat analysis showing the inferred intercellular communication network for IFNγ-IFNGR signaling in the control and the ACD cells. (**B–C**) Primary neonatal dFBs were treated with IFNγ, IL4 or IL17A for 48 hr and subjected to RNA-seq analysis. (**B**) Venn diagram comparing genes up-regulated by IFNγ, IL4. (**C**) GO pathway analysis of the genes upregulated only by IFNγ. (**D**) Primary dFBs were treated with IFNγ for 8 hr or 48 hr, and control or IFNγ treated samples were subjected to qRT-PCR analysis of *Cxcl9, Cxcl10, Stat1, Apoe,* and *Col1a1* mRNA expression (n=~3/group). (**E–H E**) Experiment scheme for collection of IFNγ-primed dFB conditioned medium (dFB^IFNγ-CM) or control dFB^ctrl-CM to stimulate naïve T cells. (**F**) ELISA analysis showing protein levels of CXCL10 in dFB^ctrl-CM and dFB^IFNγ-CM (n=3/group). (**G–H**) Naïve T lymphocytes stimulated without (**G**) or with CD3/28-Ab (**H**) were treated with dFB^ctrl-CM or dFB^IFNγ-CM with or w/o CXCR3 neutralizing antibody, and cell supernatants were collected for ELISA analysis of IFNγ protein expression (n=3/group). All error bars indicate mean ± SEM. *p<0.05, **p<0.01, ***p<0.001, ****p<0.0001, ns, non-significant.

The online version of this article includes the following figure supplement(s) for figure 5:

**Figure supplement 1.** Interaction between dFBs and T cells via the IFNG-CXCL10-CXCR3 signaling axis in ACD.

the downstream JAK-STAT1 signaling; therefore both receptor subunits are required for the functional IFNγ signaling (*Green et al., 2017*). Additionally, an unbiased IFNγ-IFNGR signaling network analysis revealed that elicitation of ACD enabled IFNγ production from CD4$^+$/CD8$^+$ or ILC1/NK T cells, acting on *IFNGR* expressed on the dFB-r5 cluster (*Figure 5A*).

To define the IFNγ-dependent immune response in dFBs and to determine how dFBs respond differently to IFNγ compared to other T cell-derived cytokines, primary dFBs were stimulated with IFNγ, IL4, or IL17A. Transcriptomic venn diagram analysis showed that IFNγ uniquely upregulated the expression of a large pool of genes, including several antiviral- or IFNγ pathway-related genes (*Figure 5B–C* and *Supplementary files 4-5*). Time course analysis showed that IFNγ treatment robustly induced the expression of *Cxcl9, Cxcl10, Stat1,* and *Ccl2* as early as 8 hr post-treatment, and suppressed the expression of lipogenesis (*Apoe*) and ECM (*Col1a1*) genes in a delayed manner (*Figure 5B* and *Figure 5—figure supplement 1C*). These in vitro changes in dFBs highly correlated with the transcriptomic change of the r5_dFB cluster upon ACD elicitation. Additionally, *Cxcr3* was specifically induced in T cells in ACD (*Figure 5—figure supplement 1D*), suggesting that IFNGR-dependent activation of dFBs may in turn promotes T cell recruitment/activation by producing CXCL9/10, which act on T cells via CXCR3.

CXCR3 is required for optimal generation of IFNG-secreting type 1T cells (*Groom et al., 2012*). Next, to determine whether IFNγ-primed dFBs can directly promote T cell activation through the CXCL9/10-CXCR3 axis, we established an in vitro co-culture system between primary dFBs and naïve lymphocytes (*Figure 5E*). ELISA confirmed that IFNγ-primed dFBs secreted high level of CXCL10 (*Figure 5F*), and conditioned medium from unstimulated control or IFNγ-primed dFBs (dFB$^{ctrl}$-CM or dFB$^{IFNγ}$-CM) was then added to primary lymphocytes with or without CD3/CD28-antibody costimulation. We found that in the absence of CD3/CD28 antibody, dFB$^{IFNγ}$-CM significantly enhanced IFNγ production from naïve T cells, and blocking CXCR3 signaling by adding a CXCR3 neutralizing antibody inhibited this effect (*Figure 5G*). Interestingly, in the presence of CD3/CD28 antibody, we found that compared to blank DMEM, dFB$^{ctrl}$-CM inhibited IFNγ expression and promoted T cell polarization towards the IL4-producing type-2 phenotype, whereas dFB$^{IFNγ}$-CM downregulated IL4 expression and restored the ability of stimulated T cells to produce IFNγ (*Figure 5H* and *Figure 5—figure supplement 1E*). In addition, the effect dFB$^{IFNγ}$-CM was partially reversed by adding a CXCR3 neutralizing antibody (*Figure 5H* and *Figure 5—figure supplement 1E*). These results indicate that IFNγ-primed dFBs are capable to shift the T1/T2 polarization balance towards type-1 effector phenotype by activating the CXCR3 pathway during T cell activation.

## Targeted deletion of *Ifngr* in dermal fibroblasts inhibited the development of type-1 skin inflammation in ACD

To validate the role of IFNGR in activating dFB, we generated tamoxifen (TAM)-inducible fibroblast-specific *Ifngr1* knockout mice by crossing *Ifngr1*$^{flox/flox}$ mice with *Pdgfra*-cre/*ERT* mice (*Xu et al., 2022*), termed as '*Ifngr1*$^{FB-iKO}$' mice (*Figure 6A*). TAM application to *Ifngr1*$^{FB-iKO}$ mice specifically ablated the expression of IFNGR1 in PDGFRA$^+$ dFBs, but not in other PDGFRA$^-$ dermal cells (*Figure 6C* and *Figure 6—figure supplement 1A and B*). As a result, DNFB-mediated development of skin eczema (*Figure 6D and E*), dermal cell infiltration (*Figure 6F*), induction of *Cxcl9/10, Cxcr3, Ifng, Cd8a* and other inflammatory genes (*Figure 6G* and *Figure 6—figure supplement 1C*), expression of CXCL9 in ICAM1$^+$ pAds (*Figure 6—figure supplement 1D–E*), and dermal infiltration of CD8$^+$ T cells (*Figure 6 H and I*) were significantly inhibited in the *Ifngr1*$^{FB-iKO}$ mice. These results demonstrate that targeted deletion of IFNγ receptor signaling in dermal fibroblasts inhibited the development of type-1 skin inflammation in ACD.

## Activation of dermal T cells and fibroblasts in human ACD skin samples

Finally, to determine whether these observations in mice are of human relevance, human ACD skin biopsies were collected for analysis. Co-immunostaining of IFNγ with CD4 and CD8 revealed that numerous CD8$^+$ IFNγ$^+$ T cells were present in the dermal micro-vascular structures, which are known to be enriched with mesenchymal stem cells (*Russell-Goldman and Murphy, 2020*), in ACD skin, but not in healthy control skin (*Figure 7A*, *Figure 7—figure supplement 1A*). In addition, within these dermal structures, IFNγ was detected exclusively from CD8$^+$ T cells (*Figure 7B*). In contrast, infiltration

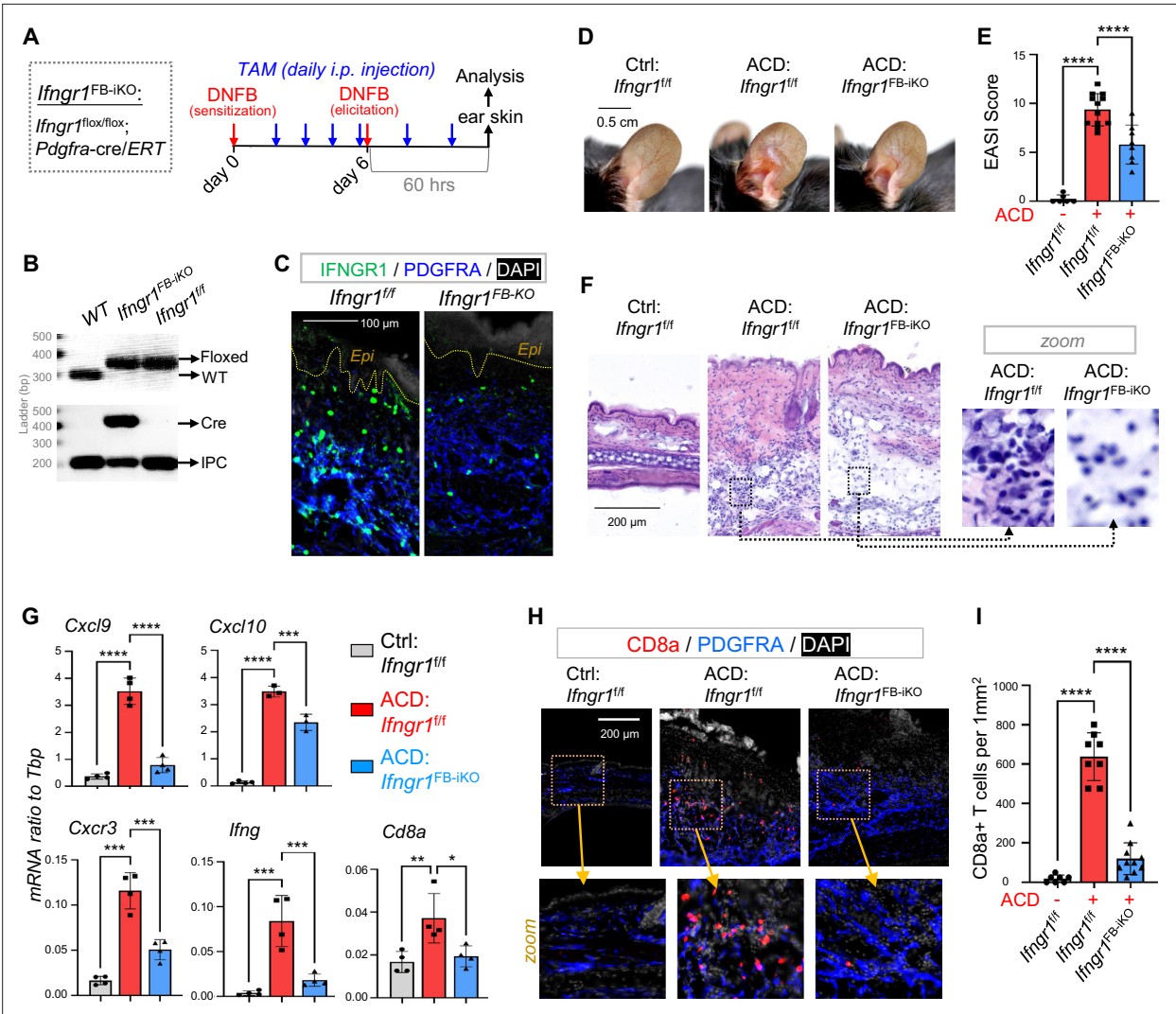

**Figure 6.** Targeted deletion of Ifngr in dFBs inhibited the development of type-1 skin inflammation in ACD. (**A**) Tamoxifen-inducible targeted deletion of *Ifngr1* in PDGFRA⁺ fibroblasts (*Ifngr1*[FB-iKO]) was achieved by crossing *Ifngr1*[f/f] and *Pdgfra*-cre/*ERT* mice, and mice were subjected to DNFB-induced ACD model as indicated. (**B**) Multiplex-PCR-based genotyping using allele specific primers yields DNA products having sizes specific for the wild-type and Ifngr1-floxed alleles. Lower gel shows gene products for Cre and/or internal product control (IPC) as indicated. (**C**) Skin samples were immunostained with IFNGR1 (green) and PDGFRA (blue), and nuclei were counterstained with DAPI (white). Scale, 100 μm. (**D–E**) Representative ear skin images (**D**) for each group at 60 hours after ACD elicitation. Br graphs (**E**) showing quantified Eczema Area and Severity Index (EASI) scores for each group (n=6–12/group). (**F**) H&E staining of skin sections for each group at 60 hours after ACD elicitation. (**G**) qRT-PCR analysis of the mRNA expression levels of indicated genes (ratios to HK gene *Tbp* were shown, n=3–4/group). (**H–I**) Immunostaining (**H**) of skin sections with anti-CD8 (red) and anti-PDGFRA (blue) antibodies, and nuclei were counter stained by DAPI (white). Zoom-in images were shown in the lower panel. Quantified results (**I**) showing the # of CD8⁺ T cells per 1 mm² of dermal area in ACD skin (n=7~9/group). All error bars indicate mean ± SEM. *p<0.05, **p<0.01, ***p<0.001, ****p<0.0001.

The online version of this article includes the following figure supplement(s) for figure 6:

**Figure supplement 1.** Targeted deletion of *Ifngr1* in dFBs inhibited the development of type-1 skin inflammation in ACD.

of CD4⁺ T cells was not observed, suggesting that CD8⁺ T cells are the major IFNγ-producing cells in the human ACD skin.

We have shown that ACD led to decrease collagen expression in dFBs (*Figure 4*). To determine how collagen was altered in human ACD, skin sections were subjected to Masson's staining to visualize collagen bundles in blue (*Figure 7C*). In line with our mouse data, collagen bundles were thinner and disorganized within the dermal micro-vascular structures where immune cells were heavily infiltrated in human ACD skin compared to healthy control skin sections (*Figure 7C*). In mouse ACD model, we showed that IFNGFR-mediated activation of dFBs promotes the infiltration and activation of CD8⁺ T

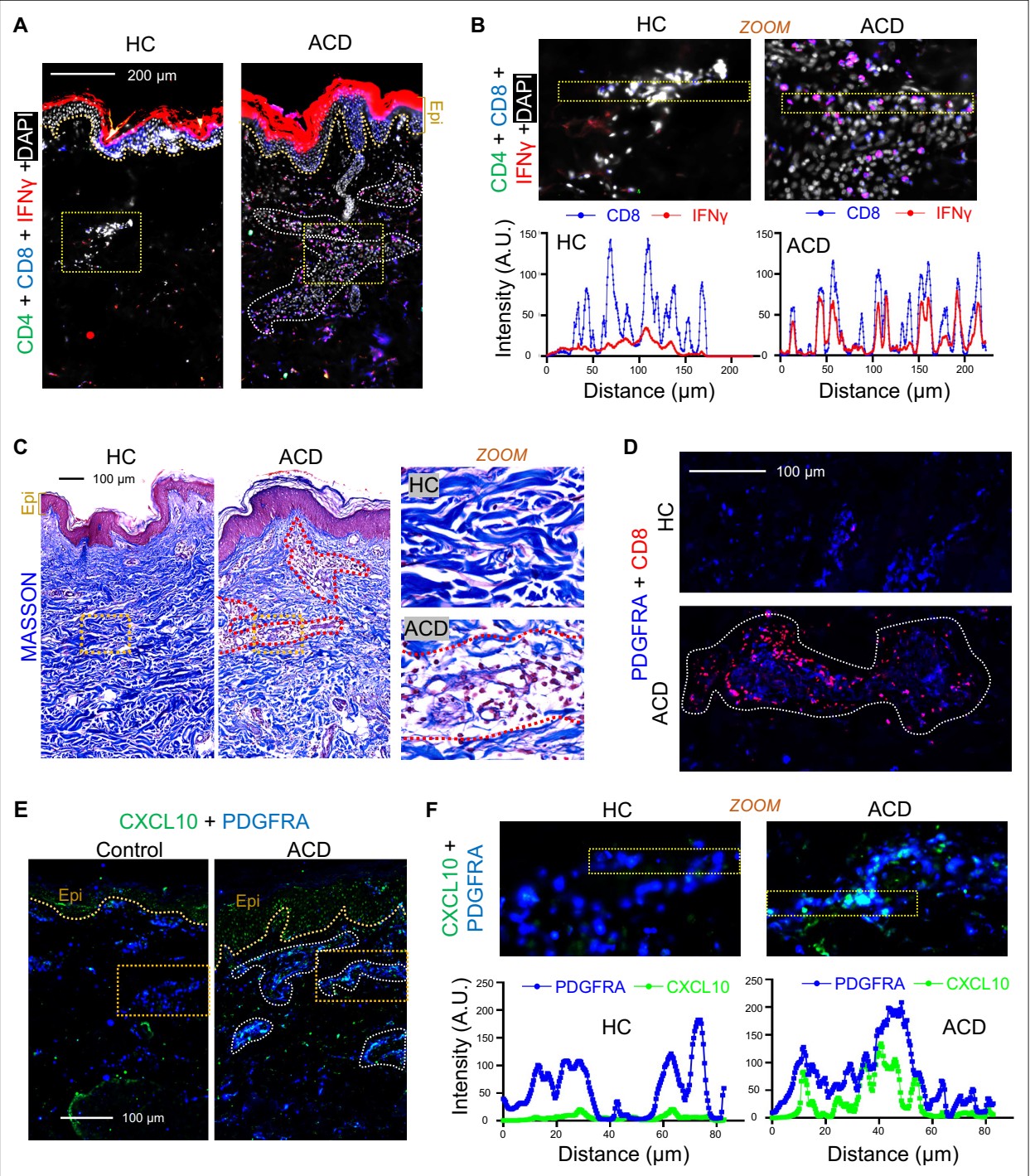

**Figure 7.** Activation of dermal T cells and fibroblasts in human ACD skin samples. (**A**) Skin sections from healthy control (HC) or ACD human skin samples were immunostained with antibodies against CD4 (green), CD8 (blue), and IFNγ (red) (representative of n=4/group). Nuclei were counter stained by DAPI (white). Scale bar, 200 μm. Zoom-in image is shown on the right-hand side. (**B**) Upper panel is the zoom-in images from panel A. Lower panel is the quantified intensity profiles of CD8 (blue) and IFNγ (red) from left to right of the images marked by yellow dashed boxes. (**C**) Skin sections from healthy control (HC) or ACD human skin samples were subjected to Masson's staining, in which collagen bundles were stained in blue. (**D**) Skin sections from healthy control (HC) or ACD human skin samples were immunostained with antibodies against CD8 (red) and PDGFRA (blue), and representative images of dermal lobular structures were shown (representative of n=4/group). (**E**) Skin sections from healthy control (HC) or ACD human skin samples were immunostained with CXCL10 (green) and PDGFRA (blue). Scale bar, 200 μm. Zoom-in image is shown on the right-hand side (**F**) Upper panel is the zoom-in images from panel E. Lower panel is the quantified intensity profiles of PDGFRA (blue) and CXCL10 (green) from left to right of the images marked by yellow dashed boxes.

*Figure 7 continued on next page*

Figure 7 continued

The online version of this article includes the following figure supplement(s) for figure 7:

**Figure supplement 1.** Activation of dermal T cells and fibroblasts in human ACD skin samples.

cells to skin dermis (*Figure 6*). Here, co-immunostaining of PDGFRA and CD8 revealed that CD8[+] T cells were infiltrated around the PDGFRA[+] dermal fibroblasts within the dermal micro-vascular structure in ACD skin (*Figure 7D*). Furthermore, PDGFRA[+] dermal fibroblasts within these micro-vascular structures expressed CXCL10 (*Figure 7E and F*, *Figure 7—figure supplement 1B*). This human dFB population correlated with the dFB-r5 cluster we identified in the murine DNFB-elicited ACD skin, in which *Cxcl9/10* were strongly induced and collagen genes were inhibited.

## Discussion

In contrast to other inflammatory skin diseases, such as atopic dermatitis (mainly involving a Th2 immune response) and psoriasis (mainly involving a Th17 immune response), ACD results from a complex but poorly defined inter-connected network between the Th1, Th2, and Th17 pathways. In this study, we provide a comprehensive single-cell analysis of the cellular interaction network of key T cell cytokines and/or chemokines in a DNFB-induced mouse model of ACD. We found that CD4[+] and CD8[+] lymphocytes are primarily polarized towards the IFNγ-producing type 1 central memory phenotype. In contrast, type 2 cytokines (IL4 and IL13) are primarily produced by basophils. Interestingly, our study identified a cluster of dermal fibroblasts with pAd signature as a dermal source for key T cell chemokines CXCL9/10. Targeted deletion of *Ifngr1* in dermal fibroblasts reduced DNFB-induced infiltration of CD8[+] T cell to the skin, the expression of *Cxcl9, Cxcl10,* and *Ifng*, and alleviated the development of ACD in mice. The results unravel a previously unrecognized role of dFBs/pAds in regulating type-1 skin inflammation.

In line with our findings, targeting type-1 immune response has emerged as effective therapeutic approach against allergic skin diseases (*Wongchang et al., 2023*; *Liu et al., 2023*; *Wu et al., 2021*). Interferon signaling, including IFNγ signaling, has been identified as a highly enriched pathway during nickel-induced contact allergy in humans (*Wisgrill et al., 2021*). Additionally, comparisons between sodium lauryl sulfate-irritated and nickel-sensitized skin samples revealed that nickel-sensitized samples were uniquely enriched with pathways related to T-cell activation, including JAK-STAT signaling, natural killer cell mediated cytotoxicity and T cell receptor signaling (*Wisgrill et al., 2021*). Furthermore, *Ifng* knockout mice failed to elicit contact hypersensitivity to urushiol (*Wakabayashi et al., 2005*), supporting that IFNγ plays a pivotal role in ACD pathogenesis.

CXCL9, CXCL10 and interferon-related pathways are recognized as potential major hallmarks that distinguish allergic-induced from irritant-induced skin inflammation (*Lefevre et al., 2021*). A study investigating the differential involvement of chemokines in chemical-induced allergic skin responses compared to irritant skin responses showed that CXCL9 and CXCL10 were among the top five chemokines that were uniquely upregulated in hapten-induced skin allergic inflammation but were absent in inflammation triggered by irritants (*Meller et al., 2007*). *Cxcl10* knockout mice display an impaired contact hypersensitivity response, characterized by reduced inflammatory cell infiltrates in the skin and reduced T cell activation and proliferative response (*Dufour et al., 2002*). CD4[+] and CD8[+] T cells showed a dose-dependent migration response to CXCL10 in vitro, and adoptive transfer of CXCL10-primed leukocytes isolated from hapten-sensitized mice increased skin inflammation after hapten challenge in vivo (*Meller et al., 2007*).

The role of dermal fibroblasts in ACD is poorly understood. Our results identified a cluster of CXCL9/10-producing dFBs/pAds in ACD, and showed that targeted deletion of *Ifngr1* in PDGFRA[+] dFB not only reduced DNFB-induced *Cxcl9/10* production but also reduced trafficking of CD8[+] T cells to affected skin sites. These results suggest that an inflammatory amplification circuit may exist between T cells and dFBs through the IFNγ–IFNGR and CXCL9/10-CXCR3 signaling network, providing a better understanding of the mechanism of action underlying IFNγ- and/or CXCL9/10-driven allergic skin inflammation. In line with our results, a recent study by Prof. Chen's group revealed the emerging roles of dFBs in the development of autoimmune diseases. They identified a subset of IFNγ-responsive dFBs which are required to recruit and activate CD8[+] cytotoxic T cells during the pathogenesis of vitiligo (*Xu et al., 2022*). In response to paracrine IFNγ signaling, dFBs orchestrate

autoimmune CD8+ T cells through secreted chemokines, including CXCL9 and CXCL10. Furthermore, the authors showed that anatomically distinct fibroblast subsets with differential IFNγ responses are the key determinants of body-level patterns of lesions in vitiligo, highlighting dFB subpopulations as therapeutic targets for vitiligo and other autoimmune diseases. While our results reveal that the cellular interaction between lymphocytes and dFBs promotes ACD pathology, a recent study shows that the innate δγ T cells interact with CD8+ T cells through the IFNγ – IFNGR pathway to restrain the proliferative and effector potential of CD8+ T cells (*Muñoz-Ruiz et al., 2023*), suggesting that activation of IFNGR signaling may either promote or inhibit type-1 effector immune response in a cell-type-specific manner.

In the murine model of ACD, the initial sensitization phase involves exposing mouse skin to a high dose of DNFB to prime memory T cells in lymphoid organs, and this is followed by a later challenge/elicitation phase, during which the mice are re-exposed to a lower dose of DNFB in a different area of the skin, distal from the original sensitization site (*Vocanson et al., 2009*; *Manresa, 2021*). Our results indicate that the type-1 inflammation observed upon ACD elicitation is predominantly driven by memory T cells recruited from lymphoid organs, rather than by skin resident memory T cells. However, other studies have shown that when the later elicitation phase is conducted on the same skin area as the initial sensitization, it results in a rapid allergen-induced skin inflammatory response, that is primarily mediated by IL17A-producing and IFNγ-producing CD8+ skin resident memory T cells (*Wongchang et al., 2023*; *Murata and Hayashi, 2020*; *Gadsbøll et al., 2020*; *Schmidt, 2017*). These studies suggest that Trm cells establish a long-lasting local memory during the initial sensitization, and upon re-exposure to the hapten in the same skin area, these site-specific Trm cells can rapidly contribute to a robust type-1 skin inflammatory response.

Infiltration of basophils into inflammatory sites has been demonstrated in several allergic diseases, including atopic dermatitis, allergic rhinitis and asthma (*Miyake and Karasuyama, 2017*) however, basophils are often regarded as minor relatives of mast cells and have long been neglected in immunological studies. Using several independent approaches, including scRNA-seq, FACS and immunostaining, we showed that basophils rapidly infiltrated the skin dermis and were the primary source of Th2 cytokines (IL4 and IL13) in DNFB-elicited ACD skin. In line with our report, a recently published scRNA-seq study of an ovalbumin (OVA)-sensitized murine skin model also unexpectedly found that mast cells/basophils, rather than T cells, are the major source of IL4 and IL13 (*Xu et al., 2022*). Another recent study showed that basophil-mediated type-2 immune response is mediated by IL3-produced by T cells in the hapten-induced ACD mouse model (*Hachem et al., 2023*). Future studies investigating cellular interactions between basophils and dFBs will advance our understanding of the basophil chemotaxis mechanism and the role of basophils in skin allergic inflammation.

Our study has a few limitations. CXCR3 is preferentially expressed on type 1 T cells and plays pivotal role in the T cell trafficking and IFNγ production (*Groom et al., 2012*). Our in vitro study is limited by only focusing on CXCL9/10-CXCR3 axis involvement in IFNγ production without studying its role in driving T cell migration. Secondly, although we have showed that the active ACD pAd (r5) and the active IFNγ-responsive vitiligo dFBs (*Xu et al., 2022*) are enriched a highly similar panel of IFNγ-inducible genes, our study is limited by the use of only one skin inflammation model. Future studies are still needed to determine whether this fibroblast-T cell axis may be broadly applied to other ACD models or to other type-1 immune response-related inflammatory skin diseases.

Taken together, our findings highlight a central role of type-1 immune response in driving ACD-like skin inflammation in both mice and human, and indicate that dermal fibroblasts are not just structural cells but also play a viscous role in amplifying type-1 inflammation in skin dermis. The interaction between T cells and dFBs may be potential therapeutic targets for allergic skin reactions. Furthermore, because IFNγ plays a central role in host defense, anti-tumor immunity, and autoimmunity (*Kwon, 2018*; *Pollard et al., 2013*), results from our study suggest that tissue-resident fibroblasts may play a role in shaping the development of type-1 immune response in other diseases, such as infectious diseases (such as herpes simplex), cancer, and autoimmune disorders, such as psoriasis, rheumatoid arthritis, and systemic lupus erythematosus.

# Materials and methods

## Animals and animal cares

ICR or C57BL/6 mice used in this study were purchased from GemPharmatech (Nanjing, China), then bred and maintained in standard pathogen free (SPF) environment of the Laboratory Animal Center in Xiamen University. C57BL/6J-Ifngr1$^{flox/flox}$ (Stock No: T009548) was originally purchased from GemPharmatech (Nanjing, China) and Pdgfra-creERT (Stock No: 018280) mice were originally purchased from Jackson laboratory (Bar Harbor, ME, USA). The tamoxifen-inducible Ifngr1 conditional knockout mice were generated by breeding Ifngr1$^{flox/flox}$ with Pdgfra-cre/ERT mice. Pdgfra-creERT; Ifngr1$^{flox/flox}$ male mice (n=4/group) were daily administrated intraperitoneally with tamoxifen (Sigma-Aldrich, St. Louis, MO, USA) solution (20 mg/mL stock solution dissolved in corn oil (Aladdin, Shanghai, China)) at 75 mg/kg body weight from post-sensitization day 3 to day 8. All mice were bred and maintained in standard pathogen free environment of the Laboratory Animal Center in Xiamen University, and all animal experiments were approved by the Animal Care and Use Committee of Xiamen University. (protocol code XMULAC20190090).

## DNFB-induced mouse model of ACD

DNFB-induced ACD mouse model was carried out according to published protocol (*Manresa, 2021*). For sensitization, the dorsal skin of 7~8 weeks old male ICR mice (n=8) was first painted with 25 μL 1.0% DNFB (dissolved in 4:1 mixture of acetone and olive oil) (Xilongs, Shenzhen, China). Six days after sensitization, to elicit ACD, 20 μL 0.2% DNFB was painted to each ear, and ear skin samples were collected at 24 hr or 60 hr post-elicitation for analysis. For treatments, 250 μL undiluted birch sap extract (BSE) was topically applied to ear skin from 12 hr post-elicitation, three times a day, and ear skin samples were collected at 60 hr post-elicitation. 1% Hydrocortisone (HC) ointment (Aveeno, Johnson & Johnson, Maidenhead, Berkshire, UK) was used as the positive group, and dH$_2$O was used as the vehicle control.

## Human skin sample collection and analysis

Fresh adult human full thickness skin biopsies, from age and sex matched healthy and ACD donors were collected by the department of dermatology at the First Affiliated Hospital of Fujian Medical University. ACD disorder was diagnosed based on their clinical appearance, history and treatment history. All ACD samples analyzed in this study were consistent with clinical and pathological criteria for ACD patients. All sample acquisitions were approved and regulated by Medical Ethics Committee of the First Affiliated Hospital of Fujian Medical University (reference number No. 2020[146]). The informed consent was obtained from all subjects prior to surgical procedures. Upon collection, these samples were directly fixed with paraformaldehyde (PFA) then proceed for paraffin embedding for histological or immunofluorescent analyses.

## Single-cell RNA library preparation, sequencing and data process

Ear skin biopsies were collected from control or DNFB-elicited skin, and skin biopsies were minced and digested with collagenase D and DNase1 to isolate single skin cells as previously described (*Sun et al., 2023*). Dead cells were removed using DeadCell Removal kit (Miltenyi Biotic,130-090-101) according to manufacturer's instruction. Live cells were counted using a hemocytometer and resuspended in 2% BSA at a concentration of 3000 cells/uL. All three samples were loaded on a 10 x Genomics GemCode Single-cell instrument that generates single-cell Gel Bead-In-EMlusion GEMs. Barcoded, full-length cDNAs were reverse-transcribed from polyadenylated mRNA. Silane magnetic beads were used to remove leftover biochemical reagents and primers from post GEM reaction mixture. Next, libraries were generated and sequenced from the cDNAs with Chromium Next GEM Single Cell 3' Reagent Kits v 2. cDNA libraries were sequenced on an Illumina Novaseq6000 platform (Illumina). The raw sequencing data was demultiplexed and aligned to the reference genome mm10-1.2.0 using Cell Ranger v3.0.2 pipiline (10 x Genomics). The generated raw gene expression matrix was converted into Seurat objects using the R package Seurat v2.0. To remove doublets and poor-quality cells, we utilized the following procedures to control for cell quality:>200 genes/cell,<5000 genes/cell;>25,000 unique molecular identifiers(UMIs); and <8% mitochondrial gene expression. Low quality cells and outliers were discarded, and only ∼ 18,241 viable cells were used for downstream analysis. These included ~9871 cells – control;~8370 cells – DNFB-elicited ACD. Unsupervised clustering and gene

expression were visualized with the Seurat 2.0 on R studio, and assignment of cell clusters was based on expression of validated marker genes. Single cell RNA libraries construction, sequencing and bioinformatic analysis was assisted by GENE DENOVO Inc (Guangzhou, China).

## Cell-chat signaling network analysis

R package CellChat 1.3.0 (*Jin et al., 2021*) was used to evaluate the potential cell-cell communication between dFB subclusters and other cell types, especially immune cells, during ACD pathogenesis. The scRNA-seq data was imported to implement CellChat platform in R software. The process includes projecting the gene expression data onto protein-protein interaction (PPI) network, then assigning a probability value to infer biological cell-cell communication network, and calculating the centrality indicator of interacting network to identify the role / contribution of each cell population in distinct signaling pathways. The number and strength of identified cell-cell communication is displayed through hierarchical graphs, circle charts, heatmaps, etc. to visualize single or multiple signaling pathways.

## Flow cytometry analysis and cell sorting

FACS procedure of dermal fibroblasts and immune cells was modified from previously established or reported methods (*Sun et al., 2023*; *Zhang et al., 2019*). Skin tissues were digested with collagenase D (Sigma-Aldrich, St. Louis, MO, USA) and Dnase 1 (Solarbio, Beijing, China) to prepare single cell suspension. Cell mixture was then filtered through 30 μm filter and treated with red blood cell lysis buffer. For analyzing cytokine expression in skin-derived T cells, isolated skin cells were first cultured in vitro in the presence of PMA (50 ng/ml) (yeasen, Shanghai, China) and Ionomycin (500 ng/ml) (Sigma-Aldrich, St. Louis, MO, USA) for 2 hr and Golgi-plug (1:1000) (BD biosciences, California, USA) for additional 1 hr prior to being subjected to FACS analysis. Briefly, freshly isolated mouse skin cells were first stained with zombie violet viability dye (BioLegend, #423114) to label dead cells. Cells were then blocked with anti-mouse CD16/32 (eBioscience, #14016185), followed by staining with an antibody cocktail for T cells containing PECy7-CD45 (BioLegend, #147704), PerCP-Cy5.5-CD8 (eBioscience, #45-0081-82), and APC-CD4 (BioLegend, #100516), an antibody cocktail for basophils and mast cells containing PECy7-CD11B (Biolegend, #101216), PE-FcεR1α (BioLegend, #134307) and APC-c-kit (BioLegend, #135108), or an antibody cocktail for immune cells containing FITC-Ly6G (eBioscience, #11593182), PE-F4/80 (eBioscience, #12480182), APC-CD11C (BioLegend, #117310), AF700-MHCII (eBioscience, #56532182), PerCP-Cy5.5-Ly6C (BioLegend, #128012), PECy7-CD11B (Biolegend, #101216), APC-Cy7-CD3 (BioLegend, #100222). To stain intracellular proteins in immune cells, stained cells were then fixed and permeabilized using the intracellular fixation and permeabilization buffer set (eBioscience, #00-8333-56) and stained by FITC-IL4 (BioLegend, #504109), FITC-IL13 (eBioscience, #53-7133-82), AF700-IFNG (BioLegend, #505824), PE-IL17A (eBioscience, #12-7177-81). FACS analysis for protein expression of each cell marker was performed by the Thermo Attune NxT machine and analyzed by FlowJo V10 software. Dead cells stained positive with zombie violet dye were excluded from the analyses.

## Histology, collagen trichrome staining, toluidine blue and immunohistochemistry (IHC)

Staining was performed on either paraffin sections or frozen sections as described previously (*Zhang et al., 2021*; *Zhang et al., 2019*). Histology was assessed by Hematoxylin and Eosin (HE) staining solutions, and collagen was stained by the Masson's Trichrome Stain Kit (Solarbio, Beijing, China). For IHC, fixed sections were permeabilized with 0.1% saponin (Sigma-Aldrich, St. Louis, MO, USA) and blocked in 5% BSA, and blocked sections were incubated with primary antibodies at 4 °C overnight followed by appropriate 488-, Cy5 or Cy3-coupled secondary antibodies in the dark for 4 hr at 4 °C. Finally, sections were mounted by ProLong Gold Antifade Mountant with DAPI (Thermo Fisher Scientific, Cleveland, OH, USA). All images were taken with Aperio VERSA Brightfield, Fluorescence Digital Pathology Scanner or Leica TCS SP8 White Light Laser Confocal Microscope, and processed by photoshop and/or Aperio ImageScope software (Leica Biosystems, Nußloch, Germany). Specific vendors, species, catalog # and dilution information for all antibodies used for IHC is listed here: rabbit anti-CXCL9/MIG (Proteintech, cat # 22355–1-AP, 1:50), rabbit anti-CXCL10/IP10 (Proteintech, cat # 10937–1-AP, 1:50), goat anti-PDGFRA (R&D system, cat #

AF1062-SP, 1:100; used for mouse skin staining), anti-IFNGR1(Abcam, cat # ab280353, 1:100), rat PE anti- CD8a (BioLegend, cat # 100708, 1:100), rabbit anti-CD4 (CST, cat # 25229 S, 1:100), Alexa Fluor 488 anti-IFN-γ (BioLegend, cat # 505813, 1:100), rat anti-PDGFRA (eBioscience, cat # 14-1401-82, 1:100; used for human skin staining), armenian hamster PE anti-FcεRIα (BioLegend, cat # 134307, 1:100), rabbit anti-TPSB (Abcam, cat # ab188766, 1:100), Alexa Fluor 488 anti-IL4 (BioLegend, cat # 504109, 1:100), Alexa Fluor 488 anti-IL13 (eBioscience, cat # 53-7133-82, 1:100), rabbit anti-pSTAT1 (CST, cat # 9167 S, 1:100), rat anti-LY6A (R&D system, cat # MAB1226, 1:100). Secondary antibodies were used: Cy3 anti-goat IgG (cat # 705-165-147), Alexa Fluor 488 anti-rabbit IgG (cat # 711-545-152), Alexa Fluor 647 anti-goat IgG (cat # 705-606-147), Alexa Fluor 647 anti-rabbit IgG (cat # 711-606-152), Alexa Fluor 647 anti-rat IgG (cat # 712-606-150). All secondary antibodies mentioned above were purchased from Jackson ImmunoResearch and used at dilution of 1:250.

## Quantitative reverse transcription-quantitative PCR (qRT-PCR) analyses

Total cellular RNA was extracted using the RNAExpress Total RNA Kit (NCM,  Suzhou, China) and 500 ng of RNA was reverse transcribed to cDNA using HiScript III Q RT SuperMix kit (Vazyme, Nanjing, China). Quantitative, realtime PCR was performed on the Qtower real time system (Analytikjena, Ilmenau OT Langewiesen • Germany) using SYBR Green Mix (Bimake, Houston, Texas, USA). All of the primers used with SYBR green were designed to span at least one exon to minimize the possibility of nonspecific amplification from the genomic DNA. The expression of *Tbp* gene (TATA-Box Binding Protein) was used as a housekeeping gene to normalize data for the expression of mouse genes. Specific primer sequences are shown in *Supplementary file 1*.

## Bulk RNA sequencing and bioinformatic analysis

Total cellular RNA were extracted using TRIzol reagent (Sigma,T9424) and RNAExpress Total RNA Kit (NCM, M050). RNA quality was analyzed by bioanalyzer and RNA samples with RIN value >7 were used for sequencing. Next generation sequencing library preparations were constructed according to the manufacturer's protocol (NEBNext Ultra RNA Library Prep Kit for Illumina). The poly(A) mRNA isolation was performed using NEBNext Poly(A) mRNA Magnetic Isolation Module (NEB). The mRNA fragmentation and priming were performed using NEBNext First Strand Synthesis Reaction Buffer and NEBNext Random Primers. First strand cDNA was synthesized using ProtoScript II Reverse Transcriptase and the second-strand cDNA was synthesized using Second Strand Synthesis Enzyme Mix. The purified double-stranded cDNA was then treated with End Prep Enzyme Mix to repair both ends and add a dA-tailing in one reaction, followed by a T-A ligation to add adaptors to both ends. Size selection of Adaptor-ligated DNA was then performed using AxyPrep Mag PCR Clean-up (Axygen), and fragments of ~420 bp (with the approximate insert size of 300 bp) were recovered. Each sample was then amplified by PCR. The PCR products were cleaned up using AxyPrep Mag PCR Clean-up (Axygen), validated using an Agilent 2100 Bioanalyzer (Agilent Technologies), and quantified by Qubit 2.0 Fluorometer (Invitrogen). Then libraries with different indexes were multiplexed and loaded on an Illumina Navoseq instrument for sequencing using a 2x150 paired-end (PE) configuration according to manufacturer's instructions. The sequences were processed and analyzed by GENEWIZ. Venn diagrams between gene sets were made by BioVenn. The top enriched genes in the active IFNγ-responsive vitiligo dFBs were from the published vitiligo study (*Xu et al., 2022*). KEGG and gene ontology (GO) pathway analysis for differentially expressed genes and correlation analysis between single cell RNA-seq with bulk RNA-seq were performed by R package clusterProfile 3.12.0.

## Cell extract preparation and ELISA assay

Skin biopsies were lysed in a lysis buffer supplemented with completed proteinase inhibitor cocktail (Apexbio, Houston, USA) to maximally preserve protein modifications as described previously (*Zhang et al., 2021*). Lysates were homogenized by sonication using digital sonifier FS- 350T (Sxsonic, Shanghai, China) followed by centrifugation to remove DNA and cell debris. Protein concentrations were measured by BCA protein assay kit. Protein lysates were subjected to enzyme-linked immunosorbent assay (ELISA) using mouse DuoSet ELISA commercial kits (R&D Systems, Minneapolis, MN, USA), following manufacturer instructions.

## Primary culture of mouse dermal fibroblasts

Primary mouse dermal fibroblasts were isolated from mouse skin as described previously (*Zhang et al., 2021*; *Zhang et al., 2019*). Isolated cells were plated on culture dish in growth medium (DMEM supplemented with 10% FBS and antibiotics/antimycotics) in a humidified incubator at 5% $CO_2$ and 37 °C under sterile conditions. Cells from the first passage were replated at $1 \times 10^5$ /mL for in vitro assays. For in vitro IFNγ treatment, dFBs were treated with either recombinant mouse IFNγ (5 ng/ml, R&D Systems, Minneapolis, MN, USA) for 8 or 24 hr before being subjected to qRT-PCR or ELISA analysis.

## Dermal fibroblasts and T cells co-culture

To collected IFNγ-primed dFB conditioned medium (CM), primary dFBs were treated with IFNγ (5 ng/ml, R&D Systems) or PBS control for 2 hr, cells were washed twice with PBS then replenished with fresh medium without IFNγ for additional 48 hr, and CM was collected for the subsequent co-culture experiment with T cells. Primary lymphocytes isolated from skin draining lymph nodes from adult C57BL/6 mice were cultured on plates precoated with anti-mouse α-CD3 (3 µg/ml, Biolegend, #100340) and anti-mouse α-CD28 (3 µg/ml, Biolegend, #102116) in RPMI medium supplemented with 10% FBS with or without dFB-CM. After 4 days, medium was collected for ELISA analysis. To neutralize CXCR3, T cells were pretreated with anti-CXCR3 neutralizing antibody (10 µg/ml, Biolegend, #126526) or control IgG antibody (10 µg/ml, Biolegend, #400902) for 30 min before adding dFB-CM. For treatments, T cells were pretreated with 5% undiluted BSE for 30 min before adding dFB-CM and 5% $dH_2O$ was used as the vehicle control.

## Statistical analysis

Experiments were repeated at least three times with similar results and were statistically analyzed by GraphPad Prism 8 software. For experiments with two groups, statistical significance was determined using Student's unpaired two-tailed t-test. For experiments with more than two groups, one-way ANOVA multiple comparison test was performed as indicated in the legend. A p value of<0.05 was considered statistically significant (*$p < 0.05$, **$p < 0.01$, ***$p < 0.001$, ****$p < 0.0001$).

## Acknowledgements

We thank the flow-cytometry and microscopic core facility at Xiamen University for aiding with flow-cytometry analysis and imaging studies.

## Additional information

### Competing interests

Xiaoting Mao, Chuan Li, Liu Hu: affiliated with Zhejiang Yangshengtang Institute of Natural Medication Co., Ltd. The author has no other competing interests to declare. Shuangping Wei: affiliated with Zhejiang Yangshengtang Institute of Natural Medication Co., Ltd. and Yang Sheng Tang (Anji) Cosmetics Co., Ltd. The author has no other competing interests to declare. The other authors declare that no competing interests exist.

### Funding

| Funder | Grant reference number | Author |
| --- | --- | --- |
| National Key Research and Development Program of China | 2023YFC2508100 | Ling-juan Zhang |
| National Natural Science Foundation of China | 82373879 | Ling-juan Zhang |
| National Key Research and Development Program of China | 2020YFA0112901 | Wenjie Liu Ling-juan Zhang |

| Funder | Grant reference number | Author |
| --- | --- | --- |
| National Natural Science Foundation of China | 81971551 | Ling-juan Zhang |
| Xiamen University Double First Class Construction Project (Biology) | DFC2024004 | Ling-juan Zhang |

The funders had no role in study design, data collection and interpretation, or the decision to submit the work for publication.

## Author contributions

Youxi Liu, Meimei Yin, Conceptualization, Data curation, Formal analysis, Validation, Investigation, Visualization, Methodology, Writing – original draft; Xiaoting Mao, Conceptualization, Data curation, Formal analysis, Investigation, Methodology; Shuai Wu, Conceptualization, Data curation, Software, Formal analysis, Methodology, Writing – original draft; Shuangping Wei, Conceptualization, Data curation, Funding acquisition, Methodology, Writing – review and editing; Shujun Heng, Zhuolin Guo, Investigation; Yichun Yang, Resources, Investigation; Jinwen Huang, Chao Ji, Resources; Chuan Li, Resources, Funding acquisition, Writing – review and editing; Liu Hu, Conceptualization, Resources, Funding acquisition, Project administration, Writing – review and editing; Wenjie Liu, Conceptualization, Resources, Data curation, Formal analysis, Supervision, Writing – original draft, Project administration, Writing – review and editing; Ling-juan Zhang, Conceptualization, Resources, Data curation, Software, Formal analysis, Supervision, Funding acquisition, Visualization, Writing – original draft, Project administration, Writing – review and editing

## Author ORCIDs

Youxi Liu ⓘ http://orcid.org/0000-0002-9084-1588
Meimei Yin ⓘ http://orcid.org/0000-0003-3718-1944
Shuai Wu ⓘ https://orcid.org/0000-0002-7569-3986
Wenjie Liu ⓘ http://orcid.org/0000-0002-8044-5338
Ling-juan Zhang ⓘ https://orcid.org/0000-0001-6937-4578

## Ethics

All sample acquisitions were approved and regulated by Medical Ethics Committee of the First Affiliated Hospital of Fujian Medical University (reference number No. 2020[146]). The informed consent was obtained from all subjects prior to surgical procedures.

All mice were bred and maintained in standard pathogen free environment of the Laboratory Animal Center in Xiamen University, and all animal experiments were approved by the Animal Care and Use Committee of Xiamen University. (protocol code XMULAC20190090).

Reviewer #1 (Public Review): https://doi.org/10.7554/eLife.94698.3.sa1
Reviewer #2 (Public Review): https://doi.org/10.7554/eLife.94698.3.sa2
Author response https://doi.org/10.7554/eLife.94698.3.sa3

# Additional files

## Supplementary files

• MDAR checklist

• Supplementary file 1. List of gene primers used for RT-qPCR.

• Supplementary file 2. List of Top 5 enriched genes in cell clusters shown in *Figure 1B–C*.

• Supplementary file 3. List of the differentially upregulated genes in the active vitiligo mouse dFBs, the ACD pAd (dFB_r5) cells, and the IFNγ-stimulated primary dFBs as shown in *Figure 5—figure supplement 1A*.

• Supplementary file 4. The RNA-seq FPKM data of primary neonatal dFBs treated with IFNγ, IL4 or IL17A.

• Supplementary file 5. List of the differentially upregulated genes in primary neonatal dFBs treated with IFNγ, IL4 or IL17A as shown in *Figure 5B–C*.

## Data availability

The accession numbers for the raw data files of the scRNA-seq analyses reported in this paper are deposited in the GEO database under accession codes: GSE224848.

The following dataset was generated:

| Author(s) | Year | Dataset title | Dataset URL | Database and Identifier |
|-----------|------|---------------|-------------|-------------------------|
| Zhang LJ | 2024 | Defining Cell Type-specific Immune Responses in Allergic Contact Dermatitis by Single-cell Transcriptomics | https://www.ncbi.nlm.nih.gov/geo/query/acc.cgi?acc=GSE224848 | NCBI Gene Expression Omnibus, GSE224848 |

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
