## [Editor Report · eLife assessment]

This **important** study uses single-cell RNA-seq to obtain a more granular understanding of cell subsets within allergic contact dermatitis in a model system with DNFB. The **convincing** data revela unique subpopulations of dermal fibroblasts as key responders to interferon gamma and likely as mediators of dermatitis. This study has many novel aspects and provides a unique resource as well.

---

## [Referee Report · Reviewer #1 (Public Review)]

In this manuscript, Liu et al. used scRNA-seq to characterize cell type-specific responses during allergic contact dermatitis (ACD) in a mouse model, specifically the hapten-induced DNFB model. Using the scRNA-seq data, they deconvolved the cell types responsible for the expression of major inflammatory cytokines such as IFNG (from CD4 and CD8 T cells), IL4/13 (from basophils), IL17A (from gd T cells), and IL1B from neutrophils and macrophages. They found the highest upregulation of a type 1 inflammatory response, centering around IFNG produced by CD4 and CD8 T cells. They further identified a subpopulation of dermal fibroblasts (pre-adipocytes found in the dermal white adipose tissue layer) that upregulate CXCL9/10 during ACD and provide functional genetic evidence in their mouse model that disrupting IFNG signaling in fibroblasts decreases CD8 T cell infiltration and overall inflammation. They identify an increase in IFNG-expressing CD8 T cells in human patient samples of ACD vs. healthy control skin and co-localization of CD8 T cells with PDGFRA+ fibroblasts, which suggests this mechanism is relevant to human ACD. This mechanism is reminiscent of recent work showing that IFNG signaling in dermal fibroblasts upregulates CXCL9/10 to recruit CD8 T cells in a mouse model of vitiligo. Overall, this is a well-presented, clear, and comprehensive manuscript. The conclusions of the study are well supported by the data, with thoughtful discussion on study limitations by the authors. One such limitation was the use of one ACD model (DNFB), which prevents an assessment of how broadly relevant this axis is. The human sample validation is limited by the multiplexing capacity of immunofluorescence markers but shows a predominance of CD8+/IFNG+ cells and PDGFRA+/CXCL10+ cells in ACD (which are virtually absent in healthy control), along with co-localization of CD8+ cells with PDGFRA+ cells. Thus, this mechanism is likely active in human ACD.

Strengths:

Through deep characterization of the in vivo ACD model using scRNA-seq, the authors were able to determine which cell types were expressing the major cytokines involved in ACD inflammation, such as IFNG, IL4/13, IL17A, and IL1B. These analyses are well-presented and thoughtful, showing first that the response is IFNG-dominant, then focusing on deeper characterization of lymphocytes, myeloid cells, and fibroblasts, which are also validated and complemented by FACS experiments using canonical markers of these cell types as well as IF staining. Crosstalk analyses from the scRNA-seq data led the authors to focus on IFNG signaling fibroblasts, and in vitro experiments demonstrate that CXCL9 and CXCL10 are expressed by fibroblasts stimulated by IFNG. In vivo functional genetic evidence demonstrates an important role for IFNG signaling in fibroblasts, as KO of Ifngr1 using Pdgfra-Cre Ifngr1 fl/fl mice, showed a reduction in inflammation and CD8 T cell recruitment. Human ACD sample staining demonstrates the likely activity of the CD8 T cell IFNG-driven fibroblast response in human disease.

Weaknesses:

The use of one model limits an understanding of how broad this fibroblast-T cell axis is during ACD. However, the authors chose the most commonly employed model and compared their data to work in a vitiligo model (another type 1 immune response) to demonstrate similar mechanisms at play. Human patient samples of ACD were co-stained with two markers at a time, demonstrating the presence of CD8+IFNG+ T cells, PDGFRA+CXCL10+ fibroblasts, and co-localization of PDGFRA+ fibroblasts and CD8+ T cells. However, no IF staining demonstrates co-expression of all 4 markers at once; thus, the human validation of co-localization of CD8+IFNG+ T cells and PDGFRA+CXCL10+ fibroblasts is ultimately indirect, although more likely than not to be true.

---

## [Referee Report · Reviewer #2 (Public Review)]

Summary: The investigators apply scRNA seq and bioinformatics to identify biomarkers associated with the DNFB-induced contact dermatitis in mice. The bioinformatics component of the study appears reasonable and may provide new insights regarding TH1 driven immune reactions in ACD in mice. However, the IF data and images of tissue sections are not clear and should be improved to validate the model.

Strengths:

The bioinformatics analysis.

Weaknesses:

The IF data presented in 4H, 6H, 7E and 7F are not convincing and need to be correlated with routine staining on histology and different IF markers for PDGFR. Some of the IF staining data demonstrates a pattern inconsistent with its target.

---

## [Author Response]

The following is the authors’ response to the original reviews.

Point-to-Point Responses to Reviewers' comments:

We extend our sincere gratitude to the reviewers for their constructive feedback and valuable suggestions, which have significantly contributed to enhancing the quality of our work. In response to the comments, we have meticulously revised our manuscript with the following updates:

(1) New Data Inclusion: We have incorporated new immunofluorescent staining images, FACS analysis of monocytes, and single-cell RNA sequencing (scRNAseq) expression analysis focusing on genes related to IFNGR, as well as T cell memory subsets (Trm, Tcm, and Tem).

(2) Comparative Analysis: We have conducted a comparative analysis between the active vitiligo dFBs and the ACD pAd (r5) identified in our study, which provides further insight into the immune response mechanisms.

(3) Discussion Expansion: We have expanded the discussion to include the role of tissue-resident memory (Trm) T cells in our model and have addressed the limitations of our animal model and in vitro studies.

(4) Supplemental Material: As requested by the reviewers, we have provided four new supplemental tables (Table S2 ~ S5) and specific information for antibodies used in our study.

Please see our Point-to-Point Responses to Reviewers' comments below:

Reviewers' comments:

**Public Reviews:**

**Reviewer #1 (Public Review):**
Summary:In this manuscript, Liu et al. used scRNA-seq to characterize cell type-specific responses during allergic contact dermatitis (ACD) in a mouse model, specifically the hapten-induced DNFB model. Using the scRNA-seq data, they deconvolved the cell types responsible for the expression of major inflammatory cytokines such as IFNG (from CD4 and CD8 T cells), IL4/13 (from basophils), IL17A (from gd T cells), and IL1B from neutrophils and macrophages. They found the highest upregulation of a type 1 inflammatory response, centering around IFNG produced by CD4 and CD8 T cells. They further identified a subpopulation of dermal fibroblasts that upregulate CXCL9/10 during ACD and provided functional genetic evidence in their mouse model that disrupting IFNG signaling to fibroblasts decreases CD8 T cell infiltration and overall inflammation. They identify an increase in IFNG-expressing CD8 T cells in human patient samples of ACD vs. healthy control skin and co-localization of CD8 T cells with PDGFRA+ fibroblasts, which suggests this mechanism is relevant to human ACD. This mechanism is reminiscent of recent work (Xu et al., Nature 2022) showing that IFNG signaling in dermal fibroblasts upregulates CXCL9/10 to recruit CD8 T cells in a mouse model of vitiligo. Overall, this is a very wellpresented, clear, and comprehensive manuscript. The conclusions of the study are mostly well supported by data, but some aspects of the work could be improved by additional clarification of the identity of the cell types shown to be involved, including the exact subpopulation discovered by scRNA-seq and the subtype of CD8 T cell involved. The study was limited by its use of one ACD model (DNFB), which prevents an assessment of how broadly relevant this axis is. The human sample validation is slightly circumstantial and limited by the multiplexing capacity of immunofluorescence markers.Strengths:Through deep characterization of the in vivo ACD model, the authors were able to determine which cell types were expressing the major cytokines involved in ACD inflammation, such as IFNG, IL4/13, IL17A, and IL1B. These analyses are well-presented and thoughtful, showing first that the response is IFNG-dominant, then focusing on deeper characterization of lymphocytes, myeloid cells, and fibroblasts, which are also validated and complemented by FACS experiments using canonical markers of these cell types as well as IF staining. Crosstalk analyses from the scRNA-seq data led the authors to focus on IFNG signaling fibroblasts, and in vitro experiments demonstrate that CXCL9 and CXCL10 are expressed by fibroblasts stimulated by IFNG. In vivo functional genetic evidence demonstrates an important role for IFNG signaling in fibroblasts, as KO of Ifngr1 using Pdgfra-Cre Ifngr1 fl/fl mice, showed a reduction in inflammation and CD8 T cell recruitment.Weaknesses:(1) The use of one model limits an understanding of how broad this fibroblast-T cell axis is during ACD. However, the authors chose the most commonly employed model and cited additional work in a vitiligo model (another type 1 immune response).

We thanks the reviewer for pointing out this limitation. Although the DNFB-elicited ACD model is the most commonly used animal model for ACD, our study is limited by the use of only one type 1 immune response model. We have now added new data (Figure 5-figure supplement 1A) showing that the active ACD pAd (r5) and the active IFNγ-responsive vitiligo dFBs (Xu et al., 2022) are enriched with a highly similar panel of IFNγ-inducible genes. Future studies are still needed to determine whether this fibroblast-T cell axis may be broadly applied to other ACD models or to other type-1 immune response-related inflammatory skin diseases.

(2) The identity of the involved fibroblasts and T cells in the mouse model is difficult to assess as scRNA-seq identified subpopulations of these cell types, but most work in the Pdgfra-Cre Ifngr1 fl/fl mice used broad markers for these cell types as opposed to matched subpopulation markers from their scRNA-seq data.

Thanks for the reviewer's constructive comments. To better showcase the dWAT layer where PDGFRA+ pAds are enriched, we have included new histological staining and PLIN1 (adipocyte marker) in new Figure 4 - figure supplement 1F-G. As shown in Figure 4 - figure supplement 1G, the PLIN1+ dWAT layer is located in the lower dermis right above the cartilage layer. In Figure 4-figure supplement 1I and J, we have shown that phosphor-STAT1 (pSTAT1), a key signaling molecule activated by IFNγ, was detected primarily in PDGFRA+Ly6A+ pAds in the lower dermis where dWAT is located. In addition, we have now included new data showing that the pAd (dFB_r5) cluster preferentially expressed the highest levels of both Ifngr1 and Ifngfr2 among all dFB subclusters (new Figure 5 - figure supplement 1B). Furthermore, we have included new co-staining data showing that CXCL9 largely co-localized with ICAM1(new Figure 4 - figure supplement 1K), a marker for committed pAds (Merrick et al., 2019), in the reticular dermis and dWAT region of the ACD skin, further confirming that CXCL9 is specifically induced in the pAd subset of dFBs. Additionally, we included new staining data showing that ACD-mediated induction of CXCL9 in ICAM1+ dFBs were largely suppressed upon targeted deletion of Ifngr1 in Pdgfra+ dFBs (new Figure 6 - figure supplement 1D-E).

(3) Human patient samples of ACD were co-stained with two markers at a time, demonstrating the presence of CD8+IFNG+ T cells, PDGFRA+CXCL10+ fibroblasts, and co-localization of PDGFRA+ fibroblasts and CD8+ T cells. However, no IF staining demonstrates co-expression of all 4 markers at once; thus, the human validation of co-localization of CD8+IFNG+ T cells and PDGFRA+CXCL10+ fibroblasts is ultimately indirect, although not a huge leap of faith. Although n=3 samples of healthy control and ACD samples are used, there is no quantification of any results to demonstrate the robustness of differences.

Thanks for the reviewer’s constructive comments. We have shown that PDGFRA colocalizes with CXCL10, in the dermal micro-vascular structures, where CD8+ T cells infiltrate around PDGFRA+ dFBs. We are sorry that due to technical issues (antibody compatibility), we cannot provide the four color co-staining as suggested by the reviewers. In order to demonstrate the robustness and reproducibility of the staining presented, we have now supplemented 4 independent images for both Fig. 7A and Fig. 7E in the updated Figure 7-figure supplement 1A-B.

**Reviewer #2 (Public Review):**
Summary:The investigators apply scRNA seq and bioinformatics to identify biomarkers associated with DNFB-induced contact dermatitis in mice. The bioinformatics component of the study appears reasonable and may provide new insights regarding TH1-driven immune reactions in ACD in mice. However, the IF data and images of tissue sections are not clear and should be improved to validate the model.Strengths:The bioinformatics analysis.Weaknesses:The IF data presented in 4H, 6H, 7E and 7F are not convincing and need to be correlated with routine staining on histology and different IF markers for PDGFR. Some of the IF staining data demonstrates a pattern inconsistent with its target.

We are sorry for the confusion, because 4H and 6H are staining on mouse skin sections, and 7E and 7F are staining on human skin sections, therefore the patterns of PDGFRA+ dFBs appeared inconsistent between species. As shown in Fig. 4H, in mouse skin, PDGFRA+CXCL9/10+ dFBs are located between the lower reticular dermis and dWAT region, where preadipocytes are located (Sun et al., 2023). To better showcase the dWAT layer where PDGFRA+ pAds are enriched, we have included new histological staining and PLIN1 (adipocyte marker) in new Figure 4 - figure supplement 1F-G. As shown in Figure 4 - figure supplement 1G, the PLIN1+ dWAT layer is located in the lower dermis right above the cartilage layer. Furthermore, we have included new co-staining data showing that CXCL9 largely co-localized with ICAM1(new Figure 4 - figure supplement 1K), a marker for committed pAds (Merrick et al., 2019), in the reticular dermis and dWAT region of the ACD skin, further confirming that CXCL9 is specifically induced in the pAd subset of dFBs.

As shown in Fig. 7E, in human skin, PDGFRA+CXCL10+ dFBs are located within the microvascular structures located at the dermal-epidermal junction (DEJ) region, where mesenchymal stem cells are enriched (Russell-Goldman & Murphy, 2020). We have included the corresponding HE histological staining image for Fig. 4H in new Figure 4-supplement 1F. Histological staining for Fig. 6H is the HE staining image in Fig. 6F. The histological staining for Fig. 7E and 7F is shown by Masson’s trichrome staining shown in Fig. 7C (a three-colour histological staining).

**Reviewer #1 (Recommendations For The Authors):**
Major comments:(1) While the focus on fibroblast and T cell interactions and overall biological findings regarding these interactions (IFNG - CXCL9/10 - CXCR3) is sound, it is slightly confusing about which exact subpopulations of these cells are involved in ACD pathogenesis as both scRNA-seq and IF are used but very broad markers are used for IF. Regarding fibroblasts, the scRNA-seq identifies the pAd (r5) cluster of fibroblasts as the main producer of CXCL9/10. However, the expression of IFNGR1 was not shown for this subpopulation as well as for other fibroblast subpopulations. Figure 6C shows IFNGR1 staining in the Ifngr1 fl/fl control mice which appears quite broad. With the seemingly broad expression of IFNGR1, why is it that only a subpopulation of fibroblasts upregulate CXCL9/10? Is there a specific location of these pAd fibroblasts that help drive this IFNG response? Please show the expression of Ifngr1 in the fibroblast scRNA-seq data.

Thanks for the reviewer’s constructive comments. We have now included new data showing that the pAd (dFB_r5) cluster preferentially expressed higher levels of both *Ifngr1* and *Ifngfr2* among all dFB subclusters (new Figure 5 - figure supplement 1B). In addition, we included new co-staining data showing that CXCL9 largely co-localized with ICAM1, a marker for committed pAds (Merrick et al., 2019), in the reticular dermis and dWAT region of the ACD skin, further confirming that CXCL9 is specifically induced in the pAd subset of dFBs.

(2) Regarding T cells, it is slightly confusing regarding what role the fibroblast-produced CXCL9/10 plays on T cell migration vs. activation. This is mainly because in vitro work focuses on T cell activation, while in vivo work seems to mainly assess T cell migration into the tissue. The in vivo studies have nicely shown that CD8 T cells are the main cell type affected by Ifngr1 iKO (i.e., a reduction of these cells), but T cell activity in vivo is not assessed (in the form of IFNG production). I have the following related questions:a. Authors do not discuss whether T cells involved in ACD in their model are tissue-resident memory T cells (Trm) or whether these are recruited from circulation. This may be possible to assess via additional analysis of the scRNA-seq data (looking for expression of Trm markers).

Thanks for the reviewer’s constructive comments. We have now included new data showing the expression of marker genes of various memory T cells in various T cell subclusters (new Figure 2 - figure supplement 1C-D). Antigen-specific CD8 or CD4 memory T cells can be classified into CD62hi/CCR7hi/CD28hi/CD27hi/CX3CR1lo central memory T cells (Tcm), CX3CR1hi/Cd28hi/Cd27lo/CD62lo/CCR7lo effector memory T cells (Tem), and CD49ahi/CD103hi/ CD69hi/BLIMP1hi tissue-resident memory T cells (Trm) *(Benichou, Gonzalez, Marino, Ayasoufi, & Valujskikh, 2017; Cheon, Son, & Sun, 2023; Mackay et al., 2013; Martin & Badovinac, 2018; Park et al., 2023)*. We observed that in ACD skin, CD4+ and CD8+ T cells predominantly expressed marker genes associated with Tcm including *Cd28*, *Cd27*, *Ccr7*, and *S1pr1/Cd62l*. In contrast, marker genes associated with Tem (*Cx3cr1*) and Trm (*Itga1/Cd49a*, *Itgae/Cd103*, *Cd69* and *Prdm1/Blimp1, Cd127/Il7r*) were only scarcely expressed in these αβ T cells, suggesting that ACD predominantly triggers a central memory T cell response in the skin.

Furthermore, this hypothesis is supported by new lymph node gene expression results. We showed that the expression of *Ifng*, but not *Il4* or *Il17a*, was rapidly induced in skin draining lymph nodes at 24 hours after ACD elicitation (new Figure 1-figure supplement 1H). This suggests a robust and systemic activation of type 1 memory T cell response in the early stage of ACD, and the migration of these lymphatic memory T cells to the skin may contribute to the exacerbation of skin inflammation.

b. Authors have focused on CXCR3 axis involvement in IFNG production (Figures 5G-H) without assessing the presumed migratory role of this axis. Presumably, CD8 T cells are recruited to the skin via the CXCL9/10-CXCR3 axis, but this would be important to clarify given other work that has demonstrated Trm involvement in ACD. Authors should at least discuss how their model and findings support, refine, or even contradict the current paradigm of Trm involvement in ACD (Lefevre et al., 2021; PMID: 34155157).

We are grateful for the constructive feedback provided by the reviewer. CXCR3 is a chemokine receptor on T cells and not only plays a pivotal role in the trafficking of type 1 T cells, but also is required for optimal generation of IFNG-secreting type 1 T cells in vivo (Groom et al., 2012). Our in vitro study is limited by only focusing on CXCL9/10-CXCR3 axis involvement in IFNγ production without studying its role in driving T cell migration. We have now addressed this limitation in the discussion section.

In the murine model of ACD, the initial sensitization phase involves exposing mouse skin to a high dose of DNFB to prime effector T cells in lymphoid organs, and this is followed by a later challenge/elicitation phase, during which the mice are re-exposed to a lower dose of DNFB in a different area of the skin, distal from the original sensitization site (Manresa, 2021; Vocanson, Hennino, Rozieres, Poyet, & Nicolas, 2009). Our updated analysis of the expression of marker genes associated with central memory T cells (Tcm), effector memory T cells (Tem), and tissue-resident memory T cells (Trm), as presented in the revised Figure 2-figure supplement 1C-D, indicates that indicate that the type-1 inflammation observed upon ACD elicitation is predominantly driven by memory T cells recruited from lymphoid organs, rather than by skin resident memory T cells. We have read the reference provided by the reviewer along with a few other related studies indicating that Trm is involved in ACD. We found that these studies performed the elicitation phase on the same skin area where the initial sensitization is conducted, and only when it results in a rapid allergen-induced skin inflammatory response, that is primarily mediated by IL17A-producing and IFNγ-producing CD8+ skin resident memory T cells (Gadsboll et al., 2020; Murata & Hayashi, 2020; Schmidt et al., 2017; Wongchang et al., 2023). These studies suggest that Trm cells establish a long-lasting local memory during the initial sensitization, and upon re-exposure to the hapten in the same skin area, these site-specific Trm cells can rapidly contribute to a robust type-1 skin inflammatory response. Therefore, a robust involvement of Trm in ACD requires a repeated exposure of the same hapten to the same skin area. We have now added related discussion in the discussion section.

c. While it may be difficult to assess given reduced numbers of CD8 T cells in the Ifngr1 iKO, is the CXCL9/10-CXCR3 axis affecting IFNG production by T cells in vivo?

Yes, we have shown in Fig. 6G that ACD-mediated induction of Ifng was significantly suppressed in the Ifngr1-iKO mice compared to the control mice.

(3) The authors cite prior work (Xu et al. Nature 2022) that demonstrated a similar mechanism for fibroblasts in recruiting vitiligo-inducing T cells. Are the pAd (r5) cluster of fibroblasts similar to the fibroblast subpopulation that drives vitiligo?

The study on mouse model of vitiligo (Xu et al. Nature 2022) did not perform single-cell RNAseq of the vitiligo mouse skin. Instead, they conducted RNAseq analysis on the sorted PDGFRA+ dFBs. Therefore, we cannot directly compare our pAd (r5) cluster with the fibroblast subpopulation that drives vitiligo. Nevertheless, by utilizing a Venn diagram to compare the top 100 lFNγ signaling dependent genes upregulated in the active vitiligo mouse dFBs and the top 100 genes enriched in our ACD pAd (dFB_r5) cells, we identified 29 commonly upregulated genes between the two conditions (Figure 5-figure supplement 1A). Furthermore, all these 29 genes were among the top IFNγ-inducible genes in primary dFBs. These shared genes include CXCL9, CXCL10, and several other downstream targets of IFNγ signaling, such as B2M, BST2, CD274, as well as the GBP family members GBP3, GBP4, GBP5, GBP7, and additional genes like H2-K1, H2-Q4, H2-Q7, H2-T23, IFIT3, ISG15, and STAT1. This result suggests that the pAd (dFB_r5) cells possess a common IFNγ-pathway gene signature with the active vitiligo mouse dFBs, indicating a potential overlap in molecular pathways.

(4) The authors should include bulk RNA-seq data from fibroblast stimulation (Figure 5b) at a minimum in the GEO submission. They should ideally include the differentially expressed genes in a supplementary table.

Thanks for the reviewer’s constructive comments. We have now included the raw FPKM file for the bulk RNAseq data shown in Fig. 5 in Supplemental Table S3, and the list for differentially expressed genes in Supplemental Table S4.

(5) The authors state that human sample stainings were n = 3 per group for healthy control and ACD (Figure 7), but no quantification or statistical testing is provided to demonstrate significant differences in findings such as co-localization of fibroblasts and T cells, IFNG+CD8+ T cells, etc.

Thanks for the reviewer’s constructive comments. We have now supplemented 4 independent images for both Fig. 7A and Fig. 7E in the new Figure 7-figure supplement 1A-B to demonstrate the robustness and reproducibility of the staining presented.

Minor comments:(1) Figure 1G, possible typos, Il14 and Il11b are on the violin plots when I believe authors meant Il4 and Il1b.

Thank a lot for pointing out these typos. We have now made the correction in the updated manuscript figure 1.

(2) The authors label cluster 27 as neutrophils based on the expression of Ly6g and S100a8. These markers are also expressed by Cd14+ inflammatory monocytes. I believe the authors need to additionally validate that these cells are neutrophils (via staining or additional analyses). Neutrophils are notoriously difficult to capture in scRNA-seq given low RNA content. Later, they are quantified by FACS using CD11b+Ly6G+ markers, but I do not believe this would distinguish them from CD14+ monocytes. As this is a relatively minor aspect of the manuscript, I consider this a minor concern, but a finding that should be as accurate as possible as Il1b is likely important, and identifying its accurate source likewise.

Thanks a lot for reviewer’s constructive comments. According to the reviewer’s suggestion, we have now added Cd14 expression in Figure 1C, and found that indeed cluster 27 express not only expressed Ly6G but also expressed Cd14. Based on literatures, the expression of Ly6G in circulating blood, spleen, and peripheral tissues is limited to neutrophils, whereas monocytes, macrophages, and lymphocytes are negative of Ly6G (Ikeda et al., 2023; Lee, Wang, Parisini, Dascher, & Nigrovic, 2013). Therefore, Ly6G can be used as a marker to distinguish neutrophils and monocytes. Although CD14 is highly expressed in monocytes, neutrophils can also express CD14 at lower level (Antal-Szalmas, Strijp, Weersink, Verhoef, & Van Kessel, 1997). Therefore, the cluster 27 is likely a mixed population of neutrophils and monocytes. So we have changed the definition of this cluster as NEU/Mon in the updated manuscript.

To confirm the presence of neutrophils and monocytes in ACD, we have included new FACS analysis of inflammatory monocytes, which are gated as CD11B+Ly6G-F4/80-CD11C-Ly6Chi, according to published FACS protocol(Rose, Misharin, & Perlman, 2012). We found that elicitation of ACD led to a transient influx of monocytes at 24 hrs post treatment, whereas the percentage of neutrophils continued to increase by 60 hours post-treatment (Figure 3L, and Figure 3-figure supplement 1G). In addition, at 60 hrs, the percentage of neutrophils (~5%) was > 10 times greater than the percentage of monocytes (~0.4%), indicating that neutrophils are the dominant granulocytes at 60 hours post ACD elicitation.

(3) The authors should include a cluster marker table as a supplementary file to accompany Figure 1C. Only top cluster markers are shown in 1C.

Thanks a lot for reviewer’s constructive comments. We have now included the top 5 enriched genes in each cell clusters shown in Fig. 1C in supplementary Table S2.

(4) Figures 2A/B have mismatched labels. There is a gdT/ILC2 label in the 2B, but not in 2A. Please match these. Along these lines, which gdT cluster is the IL17A expressing cluster as shown in 1D? Matching these labels will clarify which population is doing what.

Thanks a lot for reviewer to point out this mistake. To avoid confusion about the T cell clusters, we have added a specific recluster# for the T cell clusters as r0~r7 (Figure 2A-B). The r4 cluster is a mixed population of δγT and ILC2, therefore termed as δγT/ILC2. As shown in Figure 2-figure supplement 1E, IL17A is primarily expressed in the δγT cell (r5). We have now corrected δγT2 to δγT/ILC2 throughout the manuscript. To avoid confusion, we have now added cluster # in updated Figure 2D.

(5) In Figure 3E, the authors used CD11B as a distinguishing marker for basophils (CD11B+) vs. mast cells (CD11B-). Mcpt8 is a better distinguishing marker, so I am wondering why the authors chose CD11B.

Thanks a lot for reviewer’s comments. In scRNAseq, we did use Mcpt8 as a basophil specific marker to distinguish basophils and mast cells (see Figure 1C). However, Mcpt8 is not a surface receptor that can be used in FACS analysis. Therefore, to distinguish basophils from mast cells by FACS, we have to choose surface markers expressed on these cells. FcεR1a is a highly specific markers expressed exclusively on basophils and mast cells, and CD11B is expressed on basophils but not in mature mast cells (Hamey et al., 2021). As a result, FACS analysis of the surface expression of CD11B and FceR1a can distinguish basophils (CD11B+ FcεR1a+) from mast cells (CD11B- FcεR1a+). The use of CD11B and FcεR1a to distinguish basophils and mast cells can also been see in a published reference study (Arinobu et al., 2005).

(6) Antibody information is missing for IF studies. No clones, catalog numbers, vendors, RRIDs, or dilutions are included in the Methods section for any of the IF data.

Thanks a lot for reviewer’s constructive comments. We have now added related information for all the antibodies we used for FACS or IF data in the method section.

(7) Figure 3 supplement E and F appear to be reversed based on legend descriptions.

Thank a lot for pointing this out. We have now made the correction in the updated Supplementary file.

References:

Antal-Szalmas, P., Strijp, J. A., Weersink, A. J., Verhoef, J., & Van Kessel, K. P. (1997). Quantitation of surface CD14 on human monocytes and neutrophils. *J Leukoc Biol, 61*(6), 721-728. doi:10.1002/jlb.61.6.721

Arinobu, Y., Iwasaki, H., Gurish, M. F., Mizuno, S., Shigematsu, H., Ozawa, H., . . . Akashi, K. (2005). Developmental checkpoints of the basophil/mast cell lineages in adult murine hematopoiesis. *Proc Natl Acad Sci U S A, 102*(50), 18105-18110. doi:10.1073/pnas.0509148102

Benichou, G., Gonzalez, B., Marino, J., Ayasoufi, K., & Valujskikh, A. (2017). Role of Memory T Cells in Allograft Rejection and Tolerance. *Front Immunol, 8*, 170. doi:10.3389/fimmu.2017.00170

Cheon, I. S., Son, Y. M., & Sun, J. (2023). Tissue-resident memory T cells and lung immunopathology. *Immunol Rev, 316*(1), 63-83. doi:10.1111/imr.13201

Gadsboll, A. O., Jee, M. H., Funch, A. B., Alhede, M., Mraz, V., Weber, J. F., . . . Bonefeld, C. M. (2020). Pathogenic CD8(+) Epidermis-Resident Memory T Cells Displace Dendritic Epidermal T Cells in Allergic Dermatitis. *J Invest Dermatol, 140*(4), 806-815 e805. doi:10.1016/j.jid.2019.07.722

Groom, J. R., Richmond, J., Murooka, T. T., Sorensen, E. W., Sung, J. H., Bankert, K., . . . Luster, A. D. (2012). CXCR3 chemokine receptor-ligand interactions in the lymph node optimize CD4+ T helper 1 cell differentiation. *Immunity, 37*(6), 1091-1103. doi:10.1016/j.immuni.2012.08.016

Hamey, F. K., Lau, W. W. Y., Kucinski, I., Wang, X., Diamanti, E., Wilson, N. K., . . . Dahlin, J. S. (2021). Single-cell molecular profiling provides a high-resolution map of basophil and mast cell development. *Allergy, 76*(6), 1731-1742. doi:10.1111/all.14633

Ikeda, N., Kubota, H., Suzuki, R., Morita, M., Yoshimura, A., Osada, Y., . . . Asano, K. (2023). The early neutrophil-committed progenitors aberrantly differentiate into immunoregulatory monocytes during emergency myelopoiesis. *Cell Rep, 42*(3), 112165. doi:10.1016/j.celrep.2023.112165

Lee, P. Y., Wang, J. X., Parisini, E., Dascher, C. C., & Nigrovic, P. A. (2013). Ly6 family proteins in neutrophil biology. *J Leukoc Biol, 94*(4), 585-594. doi:10.1189/jlb.0113014

Mackay, L. K., Rahimpour, A., Ma, J. Z., Collins, N., Stock, A. T., Hafon, M. L., . . . Gebhardt, T. (2013). The developmental pathway for CD103(+)CD8+ tissue-resident memory T cells of skin. *Nat Immunol, 14*(12), 1294-1301. doi:10.1038/ni.2744

Manresa, M. C. (2021). Animal Models of Contact Dermatitis: 2,4-Dinitrofluorobenzene-Induced Contact Hypersensitivity. *Methods Mol Biol, 2223*, 87-100. doi:10.1007/978-1-0716-1001-5_7

Martin, M. D., & Badovinac, V. P. (2018). Defining Memory CD8 T Cell. *Front Immunol, 9*, 2692. doi:10.3389/fimmu.2018.02692

Merrick, D., Sakers, A., Irgebay, Z., Okada, C., Calvert, C., Morley, M. P., . . . Seale, P. (2019). Identification of a mesenchymal progenitor cell hierarchy in adipose tissue. *Science, 364*(6438). doi:10.1126/science.aav2501

Murata, A., & Hayashi, S. I. (2020). CD4(+) Resident Memory T Cells Mediate Long-Term Local Skin Immune Memory of Contact Hypersensitivity in BALB/c Mice. *Front Immunol, 11*, 775. doi:10.3389/fimmu.2020.00775

Park, S. L., Christo, S. N., Wells, A. C., Gandolfo, L. C., Zaid, A., Alexandre, Y. O., . . . Mackay, L. K. (2023). Divergent molecular networks program functionally distinct CD8(+) skin-resident memory T cells. *Science, 382*(6674), 1073-1079. doi:10.1126/science.adi8885

Rose, S., Misharin, A., & Perlman, H. (2012). A novel Ly6C/Ly6G-based strategy to analyze the mouse splenic myeloid compartment. *Cytometry A, 81*(4), 343-350. doi:10.1002/cyto.a.22012

Russell-Goldman, E., & Murphy, G. F. (2020). The Pathobiology of Skin Aging: New Insights into an Old Dilemma. *Am J Pathol, 190*(7), 1356-1369. doi:10.1016/j.ajpath.2020.03.007

Schmidt, J. D., Ahlstrom, M. G., Johansen, J. D., Dyring-Andersen, B., Agerbeck, C., Nielsen, M. M., . . . Bonefeld, C. M. (2017). Rapid allergen-induced interleukin-17 and interferon-gamma secretion by skin-resident memory CD8(+) T cells. *Contact Dermatitis, 76*(4), 218-227. doi:10.1111/cod.12715

Sun, L., Zhang, X., Wu, S., Liu, Y., Guerrero-Juarez, C. F., Liu, W., . . . Zhang, L. J. (2023). Dynamic interplay between IL-1 and WNT pathways in regulating dermal adipocyte lineage cells during skin development and wound regeneration. *Cell Rep, 42*(6), 112647. doi:10.1016/j.celrep.2023.112647

Vocanson, M., Hennino, A., Rozieres, A., Poyet, G., & Nicolas, J. F. (2009). Effector and regulatory mechanisms in allergic contact dermatitis. *Allergy, 64*(12), 1699-1714. doi:10.1111/j.1398-9995.2009.02082.x

Wongchang, T., Pluangnooch, P., Hongeng, S., Wongkajornsilp, A., Thumkeo, D., & Soontrapa, K. (2023). Inhibition of DYRK1B suppresses inflammation in allergic contact dermatitis model and Th1/Th17 immune response. *Sci Rep, 13*(1), 7058. doi:10.1038/s41598-023-34211-x

Xu, Z., Chen, D., Hu, Y., Jiang, K., Huang, H., Du, Y., . . . Chen, T. (2022). Anatomically distinct fibroblast subsets determine skin autoimmune patterns. *Nature, 601*(7891), 118-124. doi:10.1038/s41586-021-04221-8